# B38-CAP is a bacteria-derived ACE2-like enzyme that suppresses hypertension and cardiac dysfunction

Takafumi Minato[1,13], Satoru Nirasawa[2,13 ✉], Teruki Sato[1,3,13], Tomokazu Yamaguchi[1], Midori Hoshizaki[4], Tadakatsu Inagaki[5], Kazuhiko Nakahara[2], Tadashi Yoshihashi[2], Ryo Ozawa[1], Saki Yokota[6], Miyuki Natsui[1], Souichi Koyota[7], Taku Yoshiya [8], Kumiko Yoshizawa-Kumagaye[8], Satoru Motoyama[9], Takeshi Gotoh[6], Yoshikazu Nakaoka [5], Josef M. Penninger [10,11], Hiroyuki Watanabe[3], Yumiko Imai[4], Saori Takahashi [12] & Keiji Kuba [1,13 ✉]

Angiotensin-converting enzyme 2 (ACE2) is critically involved in cardiovascular physiology and pathology, and is currently clinically evaluated to treat acute lung failure. Here we show that the B38-CAP, a carboxypeptidase derived from *Paenibacillus* sp. B38, is an ACE2-like enzyme to decrease angiotensin II levels in mice. In protein 3D structure analysis, B38-CAP homolog shares structural similarity to mammalian ACE2 with low sequence identity. In vitro, recombinant B38-CAP protein catalyzed the conversion of angiotensin II to angiotensin 1–7, as well as other known ACE2 target peptides. Treatment with B38-CAP suppressed angiotensin II-induced hypertension, cardiac hypertrophy, and fibrosis in mice. Moreover, B38-CAP inhibited pressure overload-induced pathological hypertrophy, myocardial fibrosis, and cardiac dysfunction in mice. Our data identify the bacterial B38-CAP as an ACE2-like carboxypeptidase, indicating that evolution has shaped a bacterial carboxypeptidase to a human ACE2-like enzyme. Bacterial engineering could be utilized to design improved protein drugs for hypertension and heart failure.

[1] Department of Biochemistry and Metabolic Science, Akita University Graduate School of Medicine, 1-1-1 Hondo, Akita 010-8543, Japan. [2] Biological Resources and Post-harvest Division, Japan International Research Center for Agricultural Sciences, 1-1 Ohwashi, Tsukuba, Ibaraki 305-8686, Japan. [3] Department of Cardiovascular Medicine, Akita University Graduate School of Medicine, Akita, Japan. [4] Laboratory of Regulation of Intractable Infectious Diseases, National Institute of Biomedical Innovation, Health and Nutrition, 7-6-8 Saito-Asagi, Ibaraki, Osaka 567-0085, Japan. [5] Department of Vascular Physiology, Research Institute National Cerebral and Cardiovascular Center, 6-1 Kishibe Shinmachi, Suita, Osaka 564-8565, Japan. [6] Department of Materials Science, Applied Chemistry Course, Graduate School of Engineering Science, Akita University, 1-1 Tegatagakuen-machi, Akita 010-8502, Japan. [7] Molecular Medicine Laboratory, Bioscience Education and Research Support Center, Akita University, 1-1-1 Hondo, Akita 010-8543, Japan. [8] Peptide Institute, Inc., 7-2-9 Saito-Asagi, Ibaraki, Osaka 567-0085, Japan. [9] Department of Surgery, Akita University Graduate School of Medicine, 1-1-1 Hondo, Akita 010-8543, Japan. [10] IMBA -Institute of Molecular Biotechnology of the Austrian Academy of Sciences, Campus Vienna BioCenter, Vienna 1030, Austria. [11] Department of Medical Genetics, Life Science Institute, University of British Columbia, 2350 Health Sciences Mall, Vancouver, BC V6T 1Z3, Canada. [12] Akita Research Institute of Food and Brewing, 4-26 Sanuki, Arayamachi, Akita 010-1623, Japan. [13] These authors contributed equally: Takafumi Minato, Satoru Nirasawa, Teruki Sato, Keiji Kuba. ✉email: stnirasa@affrc.go.jp; kuba@med.akita-u.ac.jp

The renin–angiotensin system (RAS) has an essential role in maintaining blood pressure (BP) homeostasis, as well as fluid and salt balance[1–3]. When the RAS is activated, angiotensin-converting enzyme (ACE) cleaves the C-terminus of deca-peptide angiotensin I (Ang I or Ang 1–10) to generate a vasopressor octa-peptide angiotensin II (Ang II or Ang 1–8). ACE2 was discovered as a human homolog of ACE in 2000[4,5]. ACE2 is a negative regulator of the RAS, which catalyzes the conversion of Ang II to angiotensin 1–7 (Ang 1–7) and down-regulates Ang II levels, thereby counterbalancing ACE activity[4–6].

Physiological function of ACE2 was initially identified as a regulator of heart function and BP, and ACER, a fly homolog of ACE2, was shown to be essential for heart morphogenesis and cardiac functions in flies[7,8]. Although activation of the RAS and generation of Ang II worsen cardiovascular pathologies, such as cardiac fibrosis and pathological hypertrophy in heart failure, the enzymatic activity of ACE2 exhibits a protective role in cardio-vascular diseases[9,10]. ACE2 also has protective roles to improve the pathologies in acute respiratory distress syndrome (ARDS)/ acute lung injury and diabetic nephropathy, in which Ang II is overproduced or its signaling enhanced[11–13]. Loss of ACE2 can be detrimental, as it leads to progression of cardiac, renal, and pul-monary pathologies[11,14,15]. Treatment with recombinant human ACE2 protein (rhACE2), which is devoid of its membrane-anchored domain thus soluble, has been demonstrated to exhibit beneficial effects in various animal models including heart failure, acute lung injury, and diabetic nephropathy, and so forth[11,13,16]. rhACE2 is currently tested in the clinic to treat ARDS patients[17]. Despite its beneficial effects, rhACE2 is a glycosylated protein and thus its preparation requires time- and cost-consuming pro-tein expression system with mammalian or insect cells, which may not be advantageous in drug development and medical economy[6,18–20].

Both ACE2 and ACE proteins belong to the M2 family of zinc-binding metallopeptidases containing the HEXXH metal-coordinating motif, although the biological activities of these two enzymes are different; ACE2 functions as a mono-carboxypeptidase, whereas ACE is a dipeptidyl-carboxypeptidase[2,4,5]. Structural analyses had revealed significant homology between ACE and a carboxypeptidase from the hyperthermophilic archaeon Pfu (Pyrococcus furiosus), which is a member of the M32 family of carboxypeptidases belonging to the family of metallopeptidases with the HEXXH active-site motif[21]. The two enzymes share little amino acid sequence identity, yet they exhibit similarities in core structure and the active-site regions[21]. In addition, a structural similarity within the active-site region between ACE2 and the M32 carboxypeptidase from the bacterium Bacillus subtilis has been reported[21], suggesting that the functions might be conserved. We had previously cloned a D-aspartyl endopeptidase (paenidase I) from Paenibacillus sp. B38, a new substrain of B. subtilis[22,23]. Paenidase I cleaves D-α-Asp-containing amyloid-β peptide, which is detected in Alzheimer's disease, suggesting a potential application as a therapeutic[22].

In this study, we show that B38-CAP, a Paenibacillus sp. B38-derived carboxypeptidase, is an ACE2-like enzyme, which cleaves both Ang I and Ang II to Ang 1–7. We show that recombinant B38-CAP protein downregulates Ang II levels in mice and antagonizes Ang II-induced hypertension, pathological cardiac hypertrophy, and myocardial fibrosis. We also show beneficial effects of B38-CAP on the pathology of pressure overload-induced heart failure in mice without overt toxicities.

## Results

**Identification of B38-CAP as an ACE2-like enzyme.** To address whether there are any ACE2-like proteins in bacteria, we first searched published crystal structures of M32 carboxypeptidase in the database of MEROPS (http://merops.sanger.ac.uk/) and found that three microbial M32 carboxypeptidases are reported for their three-dimensional (3D) structures, including BS-CAP (or BsuCP) (B. subtilis), Taq (Thermus aquaticus), and Pfu (P. furiosus). Although BS-CAP, Taq, or Pfu showed low sequence identity to ACE2, we examined whether BS-CAP, Taq, or Pfu has structural similarity to human ACE2 by using the program of Molecular Operating Environment (Chemical Computing Group, Canada). The program detected only BS-CAP as a protein structurally related to ACE2 and the 3D structures of both proteins are mostly merged (Fig. 1a). Importantly, the position of key amino acids constituting the catalytic site (His-Glu-X-X-His motif) and substrate-binding region (Arg273/348, His345/234, His505/408, and Tyr515/420 in ACE2/BS-CAP) were almost identical between both proteins, implicating that BS-CAP may have similar sub-strate preference to ACE2 (Fig. 1a, Supplementary Fig. 1, and Supplementary Table 1). Indeed, a previous study on BS-CAP structure had predicted that there may be structural similarity between ACE2 and BS-CAP[24]. We further searched for bacterial proteins that exhibit high sequence identity to BS-CAP in the BLAST and found that BA-CAP, a carboxypeptidase derived from Bacillus amyloliquefaciens, is homologous to BS-CAP (Fig. 1b; Supplementary Fig. 1; Supplementary Table 1). Furthermore, we found that our recently identified bacterial strain, Paenibacillus sp. B38, also has a similar M32 carboxypeptidase to BS-CAP with high sequence identity (Fig. 1b, Supplementary Fig. 1, and Sup-plementary Table 1). We termed this Paenibacillus sp. B38-derived M32 carboxypeptidase as B38-CAP hereafter. Despite evolutionarily distant relationship to ACE2 (Fig. 1b), these bac-terial enzymes are likely homologs of ACE2 with divergent evolution.

We prepared recombinant proteins of BS-CAP, BA-CAP, and B38-CAP in the Escherichia coli protein expression system (Fig. 1c) and all of the proteins were highly expressed and soluble in E. coli and easily purified with anion-exchange and gel filtration chromatography (Supplementary Fig. 2a). Indeed, the production of recombinant B38-CAP in E. coli (16.8 mg protein yield per culture volume (L)) was more efficient in terms of the recovered protein amount compared with the production of His-tagged rhACE2 in baculovirus-Sf9 insect cells (5.42 mg protein yield per culture volume (L)). Moreover, the time for culture and purification of B38-CAP (2 days) was shorter than that of rhACE2 (6 days, not including baculovirus preparation) (Supplementary Fig. 2a–d). We first tested whether these enzymes have ACE2-like proteolytic activity to hydrolyze the fluorogenic peptide Nma-His-Pro-Lys(Dnp), which we had previously developed as a specific ACE2 substrate[19]. As a result, all the enzymes were revealed to catalyze the hydrolysis of the ACE2 substrate Nma-His-Pro-Lys (Dnp) (Fig. 1d and Table 1). When we incubated Ang II peptide with BS-CAP, BA-CAP, or B38-CAP in vitro, all of the enzymes converted Ang II to Ang 1–7 (Fig. 1e and Supplementary Fig. 3a). On the other hand, the dependency of ACE2-like enzymatic activity on anion (Cl$^-$) concentration is much higher in B38-CAP than in BS-CAP and BA-CAP (Fig. 1d), suggesting that B38-CAP is the most potent in ACE2-like activity under physiological conditions of mammals. Analysis for kinetic constants with the fluorogenic ACE2 substrate revealed that B38-CAP has the same potency as rhACE2 protein (Table 1 and Supplementary Fig. 3b). Consistently, the IC$_{50}$ values of ACE2 inhibitor (MLN-4760) for B38-CAP and human ACE2 were almost equivalent (Table 1 and Supplementary Fig. 3c). In addition, the dependence of B38-CAP proteolytic activity on pH and temperature was also similar to that of ACE2[6,19] (Supplementary Fig. 3d, e). Moreover, when various ACE2-substrate biological peptides were treated with B38-CAP, all the peptides tested were C-terminally cleaved by B38-CAP (Table 2

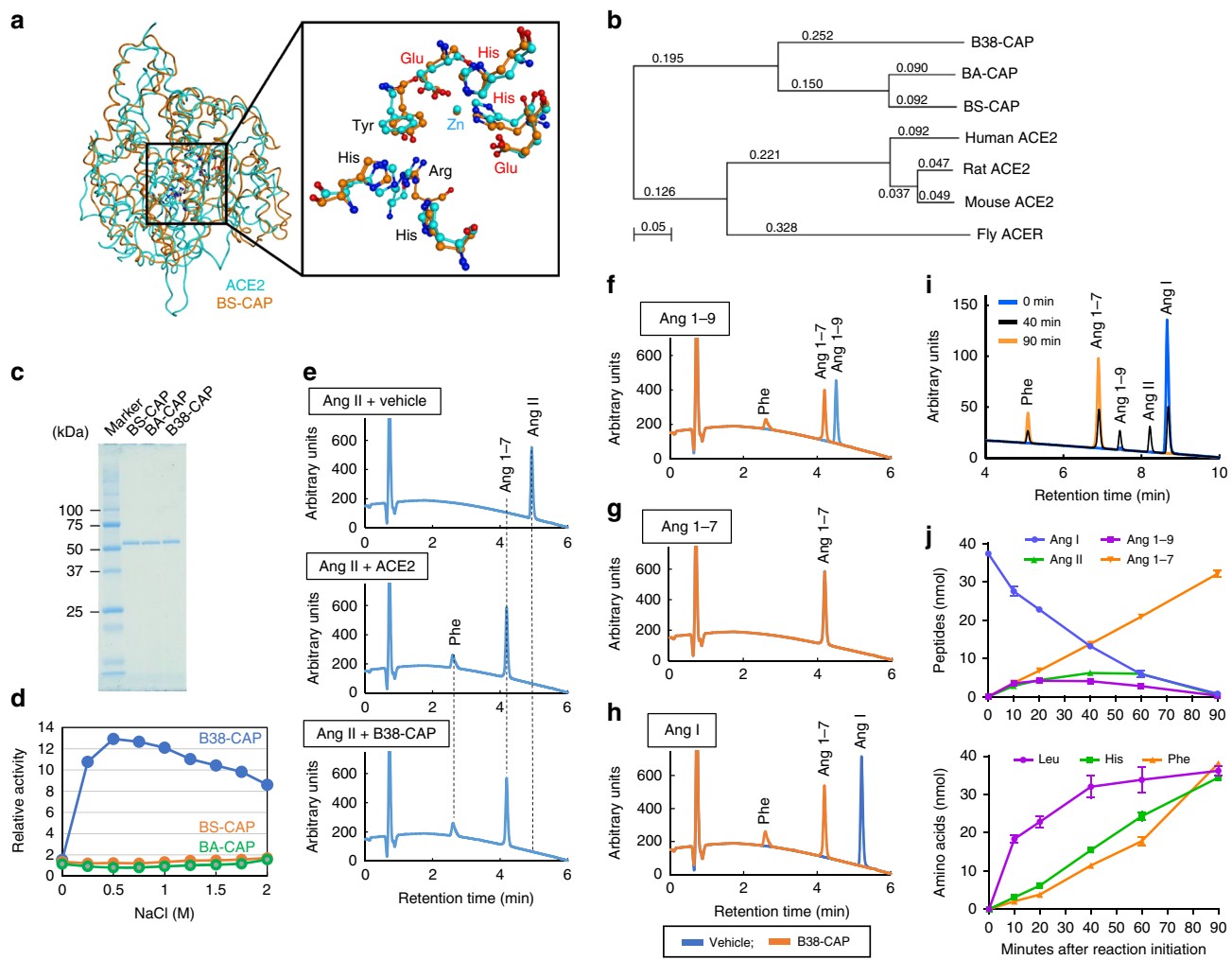

**Fig. 1 B38-CAP, a bacteria-derived carboxypeptidase, is Angiotensin-converting enzyme 2 (ACE2)-like enzyme. a** Crystal structures of BS-CAP and human ACE2 proteins. Inset: metal-coordinating residues (red) and substrate-binding residues (black) are shown. **b** Phylogenetic tree of ACE2 and bacterial ACE2-like carboxypeptidases. **c** SDS-PAGE analysis of recombinant proteins of BS-CAP, BA-CAP, and B38-CAP. **d** Dependence of ACE2-like proteolytic activity of BS-CAP, BA-CAP, and B38-CAP on anion concentration. ACE2 activity was measured with hydrolysis rate of the fluorogenic ACE2 substrate Nma-His-Pro-Lys(Dnp). **e–h** HPLC analysis of B38-CAP-treated angiotensin peptides. Ang II (**e**), Ang 1–9 (**f**), Ang 1–7 (**g**), or Ang I (**h**) (5 nmol each) was incubated with vehicle, recombinant B38-CAP protein, or recombinant ACE2 protein (5 μg each) for 90 min, then subjected to HPLC analysis. **i**, **j** Kinetic analysis for hydrolysis of Ang I with B38-CAP. HPLC analysis of angiotensin peptides generated after incubating Ang I with B38-CAP (**i**, **j**, upper panel). Amino acids in the same samples were quantified with LC-MS system (**j**, lower panel). Experiments were repeated more than three times and representative chromatography charts are shown. **j** Values are means ± SEM. $n = 3$ independent experiments.

**Table 1 Kinetic constants for hydrolysis of ACE2 substrate by B38-CAP and IC₅₀ of MLN-4760, an ACE2 inhibitor.**

|          | $K_m$ (μM)   | $k_{cat}$ (s⁻¹) | $k_{cat}/K_m$ (s⁻¹μM⁻¹) | IC₅₀ (pM)   |
|----------|--------------|-----------------|--------------------------|-------------|
| B38-CAP  | 23.3 ± 1.70  | 188 ± 6.87      | 8.07 ± 0.295             | 710 ± 173   |
| ACE2     | 23.8 ± 1.91  | 168 ± 6.18      | 7.08 ± 0.260             | 340 ± 50.1  |

and Supplementary Fig. 4a). For non-ACE2-substrate peptides Ang 1–7 and angiotensin 1–9 (Ang 1–9), however, B38-CAP did cleave Ang 1–9, whereas it did not affect Ang 1–7 (Fig. 1f, g and Table 2). Consistently, B38-CAP converted Ang I to Ang 1–7 (Fig. 1h), which is distinct from ACE2 conversion of Ang I to Ang 1–9[4]. To address how B38-CAP converts Ang I to Ang 1–7, we conducted kinetic analysis (Fig. 1i, j and Supplementary Fig. 4b). Ang 1–9, Ang II, and Ang 1–7 peptides were detectable at 10 min after mixing Ang I with B38-CAP (Fig. 1j). Ang 1–7 production

increased and finally reached the same levels of the initial Ang I amount, whereas Ang I was undetectable at 90 min (Fig. 1i, j). On the other hand, Ang 1–9 and Ang II exhibited a minor peak at 20 min and 60 min, respectively, and both peptides became undetectable at the end of the reaction period (Fig. 1j). Consistent with peptide kinetics, the amino acids leucine (Leu), histidine (His), and phenylalanine (Phe) were generated in the same order as mono-carboxyl proteolysis of Ang I, Ang 1–9, and Ang II, respectively (Fig. 1j), indicating that the conversion of Ang I to Ang 1–7 by B38-CAP is mediated through three steps of mono-carboxyl proteolysis. Therefore, B38-CAP has an ACE2-like activity, which converts both Ang II and Ang I peptides to Ang 1–7 in vitro.

**B38-CAP suppresses Ang II-induced cardiovascular pathology.** To examine the effects of B38-CAP in vivo, we first injected B38-CAP into mice and measured plasma enzymatic activity as indicative of the plasma B38-CAP levels by using a newly

**Table 2 Hydrolysis of ACE2-substrate peptides by B38-CAP.**

| Substrate | ACE2* | B38-CAP | Sequence |
|---|---|---|---|
| Angiotensin I | + | + | DRVYIHPFH↓L |
| Angiotensin 1-9 | − | + | DRVYIHPF↓H |
| Angiotensin II | + | + | DRVYIHP↓F |
| Angiotensin 1-7 | − | − | DRVYIHP |
| Apelin-13 | + | + | QRPRLSHKGPMP↓F |
| Apelin-36 | + | + | ...QRPRLSHKGPMP↓F |
| des-Arg⁹-bradykinin | + | + | RPPGFSP↓F |
| Lys-des-Arg⁹-bradykinin | + | + | KRPPGFSP↓F |
| β-Casomorphin | + | + | YPFVEP↓I |
| Dynorphin A 1-13 | + | + | YGGFLRRIRPKL↓K |
| Ghrelin | + | + | ...ESKKPPAKLQP↓R |
| Neurotensin 1-8 | + | + | pE-LYENKP↓R |

Summary of cleavability of peptides by B38-CAP. "+" indicates that it is cleaved with ACE2 or B38-CAP, whereas "−" means it is not cleaved. HPLC analyses for the metabolites of each peptide after B38-CAP treatment are shown in Supplementary Fig. 4a. *Cleavability of the peptides by ACE2 is from ref. [6].

developed B38-CAP-specific substrate Nma-Leu-Pro-Lys(Dnp). In 1 h after intraperitoneal (i.p.) injection of B38-CAP (2 mg kg⁻¹ i.p.), the concentration of B38-CAP in plasma was markedly increased and peaked (Fig. 2a). The plasma B38-CAP levels gradually decreased to almost baseline at 8 h but it was still detectable in the plasma through 12 h (Fig. 2a). Estimated initial half-life in systemic circulation was 3.5 h, which is almost similar to 1.8–8.5 h of rhACE2[25,26]. We next examined whether B38-CAP treatment affects Ang II-induced elevation of BP. We first measured the BP in the carotid artery in anesthetized mice using a transducer catheter (Fig. 2b). Intraperitoneal injection of Ang II (0.2 mg kg⁻¹ i.p.) induced acute elevation of arterial BP in wild-type mice, whereas pretreatment of B38-CAP (2 mg kg⁻¹ i.p.) significantly suppressed the Ang II-induced elevation of arterial pressure (Fig. 2c and Supplementary Fig. 5a–c). We next addressed the effects of B38-CAP on hypertension induced by chronic Ang II treatment. Although continuous infusion of Ang II with an osmotic mini-pump elevated BP as measured in conscious mice, daily i.p. injection of B38-CAP downregulated elevation of BP at days 8, 14, and 28 (Supplementary Fig. 6a–e and Supplementary Fig. 7a–e). Furthermore, we tested whether continuous infusion of B38-CAP suppresses Ang II-induced hypertension (Fig. 2d). When B38-CAP was infused subcutaneously using an osmotic mini-pump, B38-CAP was detectable in the plasma of mice for 14 days (Supplementary Fig. 8a, b). Although it had been reported that an immune response is associated with the chronic infusion of rhACE2 resulting in the degradation of rhACE2[26], this was not observed for B38-CAP; there were no antibodies against B38-CAP detectable in the serum of mice infused with B38-CAP for 2 weeks (Supplementary Fig. 8c). Implantation of B38-CAP-filled osmotic mini-pumps significantly suppressed Ang II-induced hypertension in conscious mice (Fig. 2e–g) without affecting the heart rate (Fig. 2h). These results indicate that B38-CAP antagonizes the vasopressor effect of Ang II.

We further examined the effects of B38-CAP treatment on Ang II levels in the blood. In the acute experiment with i.p. injection of Ang II (Fig. 2b), pretreatment of B38-CAP markedly down-regulated a massive increase of plasma Ang II levels at 5 min after Ang II injection (Fig. 2i). Consistently, in the chronic experiment with continuous infusion of Ang II (Fig. 2d), continuous infusion of B38-CAP with additional osmotic pump significantly decreased Ang II levels in the plasma at day 14 (Fig. 2j). As Ang 1–7 is known to exert beneficial effects in the cardiovascular systems through Mas/Ang 1–7 receptor[27], we also measured Ang

1–7 levels in the plasma of these mice. Ang 1–7 levels in the plasma were significantly upregulated in both acute and chronic experimental settings (Fig. 2k, l). On the other hand, when we co-treated the Ang II-injected mice with B38-CAP and A779, an antagonist for Mas/Ang 1–7 receptor, the suppressive effects of B38-CAP on Ang II-induced elevation of BP were not affected (Fig. 2m and Supplementary Fig. 9a–d). Furthermore, co-treatment of Ang 1–7 and Ang II did not downregulate Ang II-induced elevation of BP (Supplementary Fig. 9e–h). Thus, the hypotensive effects of B38-CAP are primarily mediated through downregulation of Ang II levels.

As chronic infusion of Ang II induces cardiac hypertrophy and fibrosis, we examined the hearts of mice chronically treated with high-dose Ang II (1.5 mg kg⁻¹ per day) with or without B38-CAP (3 mg kg⁻¹ per day) for 2 weeks. B38-CAP suppressed Ang II-induced cardiac hypertrophy and increase of heart weight (HW) as measured with HW-to-body weight ratios (HW/BW) or HW to tibia length (HW/TL) (Fig. 3a–c and Supplementary Table 2). Consistently, Ang II-induced wall thickening of the hearts was significantly downregulated by B38-CAP treatment as shown by echocardiography (Fig. 3d, e and Supplementary Table 2). In addition, mild decrease of cardiac contractility in Ang II-infused mice were prevented by B38-CAP treatment as determined by % fractional shortening (%FS) (Fig. 3f and Supplementary Table 2). Moreover, B38-CAP significantly prevented Ang II-induced cardiac fibrosis (Fig. 3g, h) and upregulation of fibrotic genes expression (Collagen 8a (Col8a1), Periostin (Postn), and TGF-β2 (Tgfb2)) (Fig. 3i–k). Similar results were obtained when B38-CAP was daily i.p. injected for 4 weeks to Ang II-infused mice (Supplementary Fig. 7f–n and Supplementary Table 3). These results demonstrate that B38-CAP suppresses Ang II-induced hypertension, cardiac hypertrophy, and fibrosis.

**B38-CAP mitigates pressure overload-induced heart failure.**
We further treated the mice under pressure overload cardiac stress induced by transverse aortic constriction (TAC) with continuous infusion of B38-CAP (2 mg kg⁻¹ per day), which was initiated immediately after TAC surgery (Fig. 4a). After 2 weeks of TAC, the HW (HW/BW or HW/TL) was also significantly decreased in the B38-CAP-treated group as compared with vehicle-treated controls (Fig. 4b–d). In addition, pulmonary congestion was suppressed by B38-CAP as determined by lung weight to BW ratio (LW/BW) and LW/TL ratio (Fig. 4e, f). Echocardiography also showed that wall thickening was significantly downregulated by B38-CAP treatment (Fig. 4g–i and Supplementary Table 4). In addition, although %FS was significantly decreased in the vehicle treatment group, %FS was preserved in B38-CAP-treated mice (Fig. 4g–l). Similarly, B38-CAP treatment significantly suppressed the increased expression of mRNA associated with cardiac hypertrophy, such as ANF (atrial natriuretic factor), BNP (brain natriuretic peptide), and β-myhc (β-myosin heavy chain, Myh7) in the TAC mice (Fig. 5c–e).

Histological analysis further revealed that B38-CAP treatment reduced the area of cardiac fibrosis in the interstitial space and perivascular region in the hearts of TAC mice (Fig. 5a, b). Consistently, although the expression of the pro-fibrotic genes Col8a1, Postn, and Tgfb2 were increased in the hearts of vehicle-treated mice with TAC, B38-CAP markedly downregulated the expression of those pro-fibrotic genes (Fig. 5f–h). These results indicate that exogenous B38-CAP treatment protects mice from pressure overload-induced cardiac dysfunction, hypertrophy, and fibrosis. Furthermore, we examined whether any potential toxic effects of B38-CAP on the liver and kidneys, and both serum markers of liver injury and kidney dysfunction, were not affected by B38-CAP, as measured with aspartate transaminase (AST) or

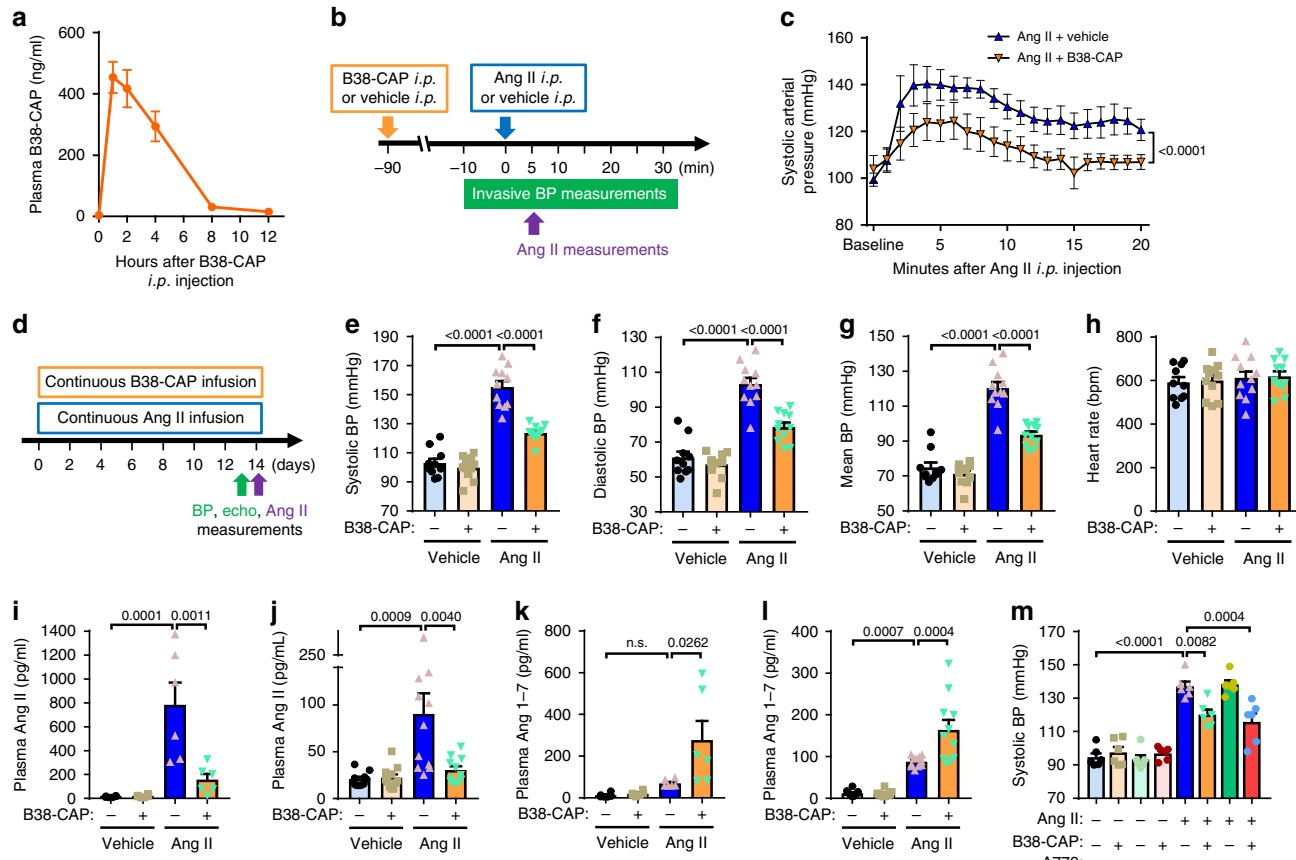

**Fig. 2 Effects of B38-CAP on plasma angiotensin II levels and blood pressure in mice. a** Plasma B38-CAP levels in mice after intraperitoneal injection of B38-CAP (2 mg kg$^{-1}$). B38-CAP activity was measured with the B38-CAP substrate Nma-Leu-Pro-Lys(Dnp) ($n = 4, 6, 8, 6, 6,$ and 6 for 0, 1, 2, 4, 8, and 12 h, respectively). **b, c** Invasive measurements of arterial blood pressure (BP). Experimental protocol (**b**); mice pretreated with B38-CAP (2 mg kg$^{-1}$ i.p.) had liquid-filled catheter inserted into carotid artery for BP measurements and Ang II (0.2 mg kg$^{-1}$ i.p.) was injected and BP measured. Systolic arterial pressure is shown (**c**). ($n = 6$ mice per group). **d–h** Blood pressure measurements with conscious mice. Experimental protocol (**d**); mice were treated with continuous infusion of vehicle, Ang II (1.5 mg kg$^{-1}$ per day), B38-CAP (3 mg kg$^{-1}$ per day), or Ang II (1.5 mg kg$^{-1}$ per day) plus B38-CAP (3 mg kg$^{-1}$ per day), and BP was measured by tail-cuff system after 2 weeks. Systolic (**e**), diastolic (**f**), and mean (**g**) BP and heart rate (**h**) are shown for mice treated vehicle + vehicle ($n = 10$), vehicle + B38-CAP ($n = 11$), Ang II + vehicle ($n = 11$), and Ang II + B38-CAP ($n = 11$). **i–l** Measurements of Ang II and Ang 1–7 in the plasma of mice. The plasma was obtained from the mice treated acutely (**i, k**) and chronically (**j, l**) with Ang II, in the cohort of **b, c**, and **d–h**, respectively. Ang II (**i, j**) and Ang 1–7 (**k, l**) levels were measured with ELISA. **m** BP measurements in conscious mice. The mice pretreated with B38-CAP (2 mg/kg i.p.) at 90 min before acquisition of baseline BP were treated with Ang II (0.2 mg kg$^{-1}$ i.p.), with or without A779 (0.2 mg kg$^{-1}$ i.p.). BP was measured every 5 min by tail-cuff system (Supplementary Fig. 9a–d) and the BP at 5 min after the last injection is shown ($n = 6$ mice per group). All values are means ± SEM. **c, e–m**, Two-way ANOVA with Sidak's multiple-comparisons test. Numbers above square brackets show significant *P*-values.

alanine transaminase (ALT) and blood urea nitrogen (BUN) or Creatinine (Cr), respectively (Fig. 6a–d). In addition, the decrease of BW due to TAC heart failure was prevented by B38-CAP treatment (Supplementary Table 4). These results suggest that B38-CAP does not exhibit overt side effects in mice for at least 2 weeks after treatment.

**B38-CAP improves established hypertension and heart failure.**
To determine therapeutic effects of B38-CAP in established diseases, we first examined whether B38-CAP treatment improve established hypertension in the mice, which have received Ang II infusion prior to the treatment (Fig. 7a). When BP was elevated after 7 days of Ang II infusion, daily i.p. injection of B38-CAP was initiated (Fig. 7a). B38-CAP treatment significantly downregulated Ang II-induced increase of BP to the levels in vehicle-treated control mice (Fig. 7b–e). Thus, B38-CAP ameliorated established hypertension. Next, we investigated the effects of B38-CAP on established cardiac dysfunction. At 5 weeks after TAC surgery, the mice (C57BL/6J background) exhibits cardiac

dysfunction with significant decrease of %FS (Fig. 7f, g). The osmotic mini-pumps containing B38-CAP or vehicle were implanted at this time point and B38-CAP (2 mg kg$^{-1}$ per day) was infused for 2 weeks. Treatment with B38-CAP significantly increased %FS as compared with that in vehicle-treated mice or in the mice before treatment (Fig. 7g and Table 3), indicating that B38-CAP improved cardiac dysfunction. In addition, cardiac hypertrophy was significantly suppressed by B38-CAP treatment (Fig. 7h–j) and pulmonary congestion was also improved by B38-CAP (Fig. 4k, l). Consistently, B38-CAP treatment significantly downregulated increased expression of mRNA associated with the pathology of cardiac hypertrophy (*BNP* and *β-myhc*) and fibrosis (*Col8a1*, *Postn*, and *Tgfb2*) (Fig. 7m–q). Furthermore, we challenged B38-CAP to severe cardiac dysfunction in C57BL/6N mice under TAC, in which the mice exhibit profound decline of cardiac contractility at 2 weeks after TAC (Fig. 4l and Supplementary Fig. 10a–c). When B38-CAP treatment was started at 2 weeks after TAC, B38-CAP markedly suppressed progression of heart failure in C57BL/6N mice (Supplementary Fig. 10b–o and

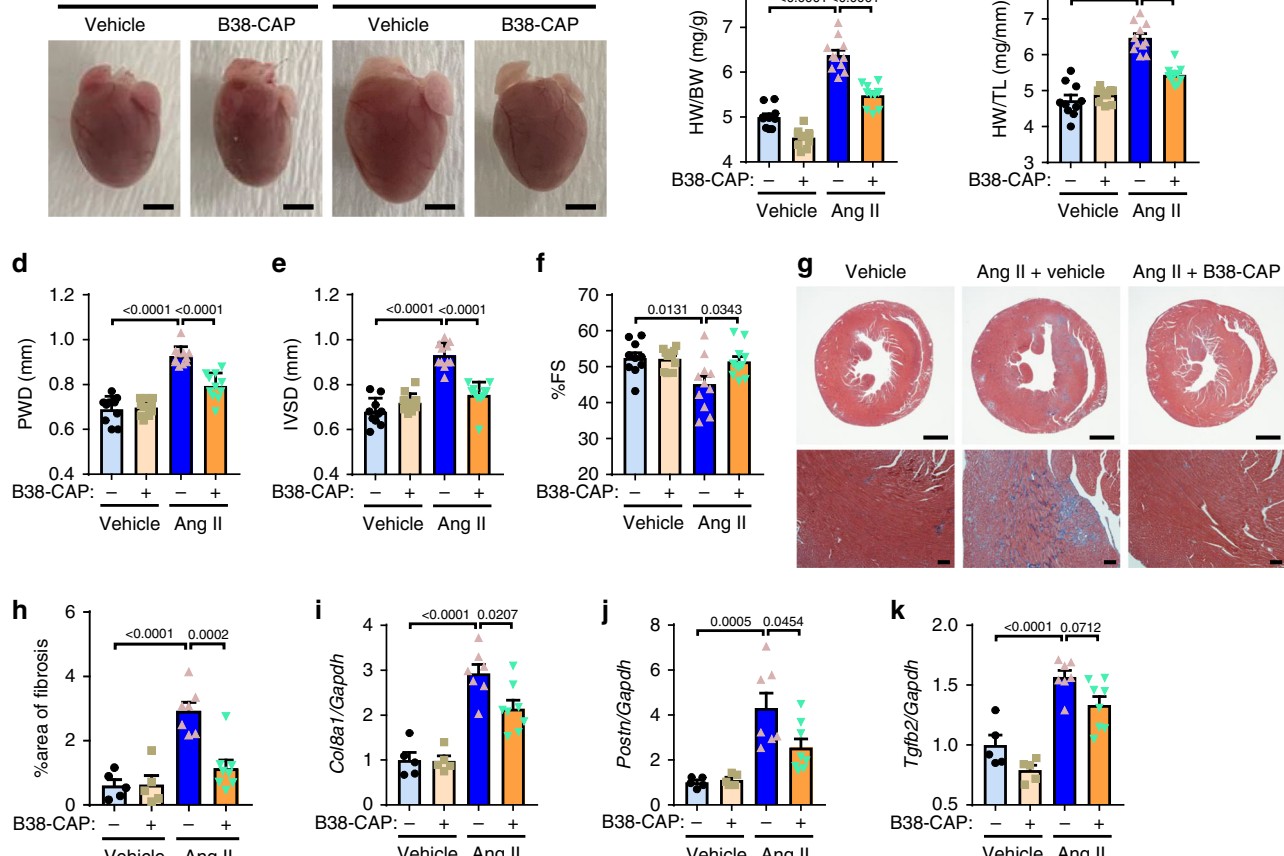

**Fig. 3 Effects of B38-CAP on angioteinsin II-induced cardiac hypertrophy and fibrosis. a–c** Cardiac hypertrophy in the mice chronically co-treated with Ang II (1.5 mg kg$^{-1}$ per day) and B38-CAP (3 mg kg$^{-1}$ per day) in the cohort of Fig. 2d–h. Macroscopic heart images (**a**), heart weight to body weight ratio (HW/BW) (**b**), and heart weight to tibia length ratio (HW/TL) (**c**) in mice treated vehicle + vehicle (n = 10), vehicle + B38-CAP (n = 11), Ang II + vehicle (n = 11), and Ang II + B38-CAP (n = 11). Bars indicate 2 mm. **d–f** Echocardiography parameters of left ventricular end-diastolic posterior wall thickness (PWD) (**d**), end-diastolic interventricular septal wall thickness (IVSD) (**e**), and %fractional shortening (%FS) (**f**) in the mouse hearts. Complete echocardiography data are shown in Supplementary Table 2. **g, h** Histology of hearts. Masson's trichrome staining (**g**); bars indicate 2 mm and 100 μm in the upper panels and lower panels, respectively. Quantification of fibrosis in the hearts (**h**) of mice treated with vehicle + vehicle (n = 5), vehicle + B38-CAP (n = 5), Ang II + vehicle (n = 7), and Ang II + B38-CAP (n = 8). **i–k** qRT-PCR analysis of pro-fibrotic gene expressions in the hearts of mice treated with vehicle + vehicle (n = 5), vehicle + B38-CAP (n = 5), Ang II + vehicle (n = 7), and Ang II + B38-CAP (n = 8); mRNA levels of Collagen 8a (*Col8a1*) (**f**), Periostin (*Postn*) (**g**), and TGF-β (*Tgfb2*) (**h**) normalized with *Gapdh*. All values are means ± SEM. **b–f**, **h–k** Two-way ANOVA with Sidak's multiple-comparisons test. Numbers above square brackets show significant *P*-values.

Supplementary Table 5), albeit mainly through decrease of LV dimensions rather than wall thickness. These results indicate that B38-CAP exerts therapeutic effects in established cardiac dysfunction.

## Discussion
In this study, we elucidated that bacteria-derived carboxypeptidases have ACE2-like enzymatic activity and showed that B38-CAP cleaves Ang II and Ang I to Ang 1–7 and downregulates Ang II levels in mice. We demonstrated that Ang II-induced hypertension, cardiac hypertrophy, and fibrosis were suppressed by B38-CAP treatment. We further showed that B38-CAP improves cardiac dysfunction, hypertrophy, and fibrosis induced by pressure overload in mice.

Among three bacterial carboxypeptidases we tested, only B38-CAP showed dependence of proteolytic activity on anion concentration, which is characteristic of ACE2 activity[6]. B38-CAP also showed pH optimum of 7.5 equivalent to rhACE2[6,19]. In addition, IC$_{50}$ of MLN-4760 was also equivalent between rhACE2 and B38-CAP. Although B38-CAP exhibited quite

similar proteolytic activity to rhACE2, there seems a difference in substrate specificity between two enzymes. B38-CAP is likely to have more broad specificity for proteolytic effects on peptides, because B38-CAP converted Ang I and Ang 1–9 into Ang 1–7, whereas ACE2 does not cleave Ang 1–9. The superimposition of ACE2 and BS-CAP structures indicated that the positions of substrate-binding amino acid residues and metal-binding residues are matched. Although the ACE2 substrate Nma-His-Pro-Lys (Dnp) was cleaved by B38-CAP with the same potency as ACE2, we found that the Nma-Leu-Pro-Lys(Dnp) was catalyzed specifically by B38-CAP but not by ACE2 (not shown), suggesting that the S2-subsite of B38-CAP is more hydrophobic than ACE2. The difference in substrate specificity of B38-CAP and ACE2 should be further elucidated by our ongoing analysis for crystal structure of B38-CAP.

Although the hypotensive action of B38-CAP is mediated mainly through Ang II downregulation, B38-CAP (2 ~ 3 mg kg$^{-1}$ per day) exhibits a more potent therapeutic effect in TAC-induced heart failure than in Ang II-induced heart failure (e.g., 87.9% inhibition of cardiac hypertrophy (HW/BW increase) in

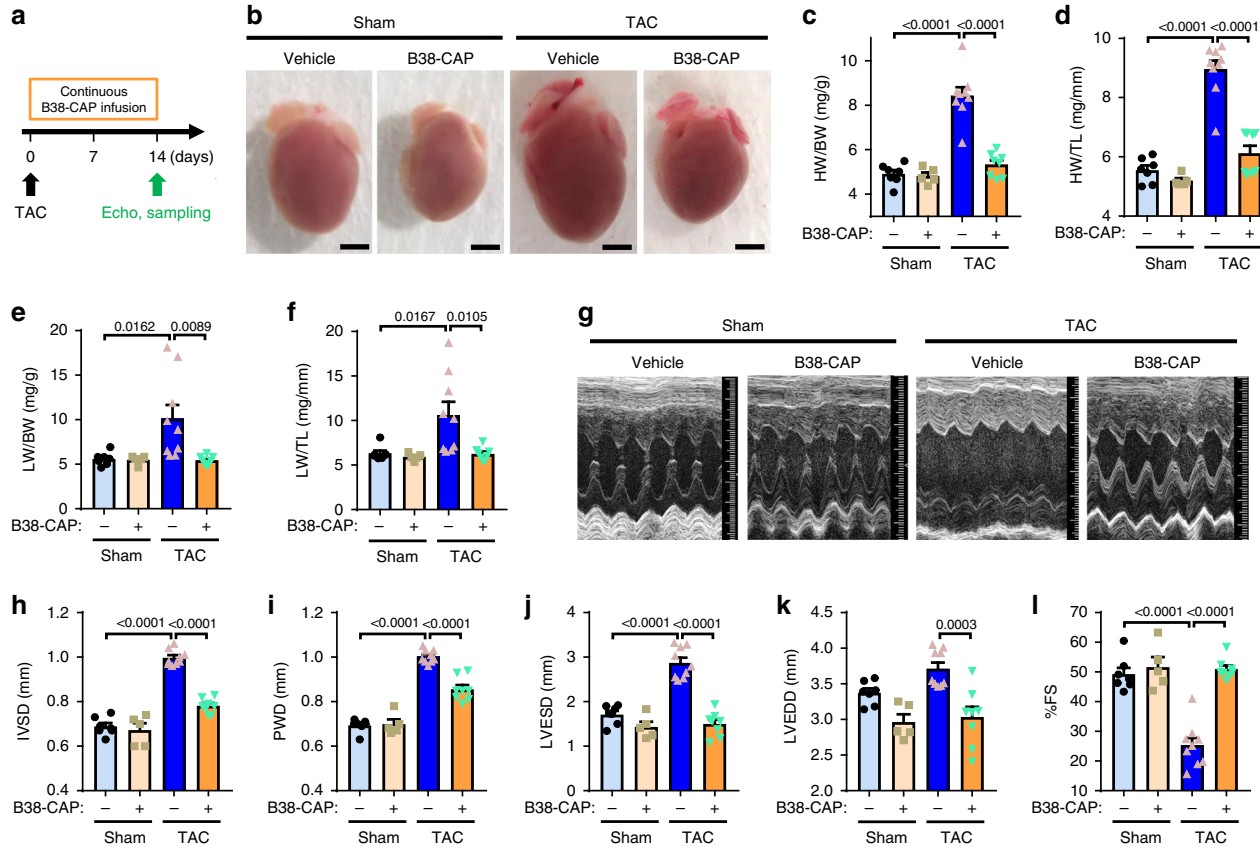

**Fig. 4 B38-CAP mitigates pressure overload (TAC)-induced cardiac dysfunction and hypertrophy. a** Experimental protocol. The mice were subjected to the surgery of transverse aortic constriction (TAC) surgery and then continuous infusion of B38-CAP (2 mg kg$^{-1}$ per day) was initiated. **b–f** B38-CAP suppressed cardiac hypertrophy. Representative photograph (**b**) of the hearts of mice under TAC. Bars indicate 2 mm. HW/BW (**c**), HW/TL (**d**), lung weight to body weight ratio (LW/BW) (**e**), and lung weight to tibia length ratio (LW/TL) (**f**) in the mice treated with sham + vehicle ($n = 7$), sham + B38-CAP ($n = 5$), TAC + vehicle ($n = 9$), and TAC + B38-CAP ($n = 8$). **g–l** Echocardiography measurements. Representative M-mode echocardiography images (**g**), measurements of IVSD (**h**), PWD (**i**), LVESD (**j**), LVEDD (**k**), and %FS (**l**) are shown. Complete echocardiography data are shown in Supplementary Table 4. All values are means ± SEM. **c–f**, **h–l** One-way ANOVA with Sidak's multiple-comparisons test. Numbers above square brackets show significant P-values.

TAC model vs. 53.2% inhibition in Ang II-infusion model). On the other hand, in a previous study, rhACE2 showed similar anti-hypertrophic effects in both TAC and Ang II-infusion models[16]. The difference may be explained by the slight difference of substrate specificity of B38-CAP and ACE2. Although ACE2 converts Ang I to Ang 1–9 inefficiently and requires ACE for further conversion of Ang 1–9 to Ang 1–7[6], B38-CAP targets all the Ang I, Ang 1–9, and Ang II peptides as a mono-carboxypeptidase. Thus, the conversion of Ang I and Ang II to Ang 1–7 by B38-CAP may contribute to more efficient down-modulation of RAS in the TAC heart failure. In addition, Ang 1–7 is a vasoprotective peptide which acts through its cognate Mas receptor, and also has anti-fibrotic and cardioprotective functions in heart failure[10]. Thus, B38-CAP-mediated degradation of Ang I into Ang 1–7 would be beneficial in enhancing Ang 1–7 generation for treating failing hearts. Furthermore, as ACE2 and B38-CAP target other biological peptides than angiotensin peptides, subtle differences in substrate specificity may license B38-CAP to degrade such peptides in a different manner or even to have a new peptide substrate different from ACE2 targets.

In addition to the currently used drugs to inhibit Ang II generation or signaling, such as ACE inhibitors or Angiotensin receptor blockers, direct down-modulation of Ang II levels by rhACE2 protein is one of the promising candidates for new therapeutic strategy in cardiovascular disease and other Ang II-related diseases, e.g. ARDS. On the other hand, although mass production of rhACE2 as a protein drug costs due to requirement of mammalian cell expression systems, B38-CAP is easily prepared with *E. coli* expression system and is cost effective. Therapeutic efficacy and less toxicity of B38-CAP in mouse heart failure models would warrant further investigation of B38-CAP or other microbial carboxypeptidases in disease models. Furthermore, human ACE2-like enzyme in bacteria might pave the way to a new strategy to engineer evolution of bacterial proteins for better designing and preparations of recombinant protein drugs.

## Methods

**Searching for bacteria-derived ACE2-like enzymes.** Sequence comparison was performed using BLAST and MEROPS (http://merops.sanger.ac.uk/) tools available online. Similarity search and superposition of a 3D structure of proteins was performed by Molecular Operating Environment (MOE 2016.08; Chemical Computing Group, Inc., Montreal, QC, Canada). Multiple sequence alignment between enzyme sequences was performed using the CLUSTALW tool available online. Phylogenetic tree drawing was performed using NJplot program (http://doua.prabi.fr/software/njplot).

**Recombinant proteins.** Genomic DNAs were isolated from *Paenibacillus* sp. B38[22], *B. subtilis* subsp. *subtilis* NBRC 13719, and *B. amyloliquefaciens* NBRC 3022 (Supplementary Table 1). Expression plasmids, which encode B38-CAP, BS-CAP,

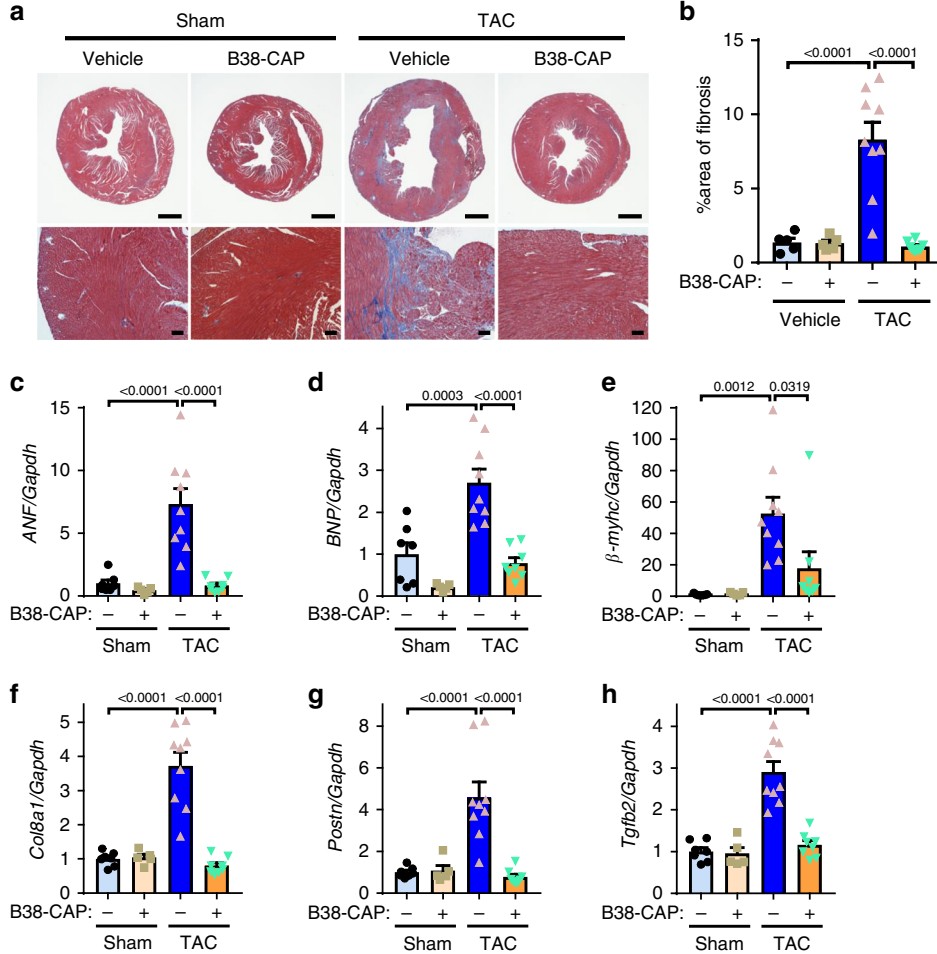

**Fig. 5 B38-CAP suppresses TAC-induced cardiac fibrosis. a**, **b** Histology. The hearts of B38-CAP or vehicle-treated mice under TAC were stained with Masson's trichrome. Bars indicate 1 mm (upper) or 100 μm (lower). **c–h** qRT-PCR analysis for the expression of heart failure genes and pro-fibrosis genes; mRNA levels of atrial natriuretic factor (*ANF*) (**c**), B-type natriuretic peptide (*BNP*) (**d**), β-myosin heavy chain (*β-myhc*) (**e**), Collagen 8a (*Col8a1*) (**f**), Periostin (*Postn*) (**g**), and TGF-β (*Tgfb2*) (**h**) in the hearts of mice treated with sham + vehicle (*n* = 7), sham + B38-CAP (*n* = 5), TAC + vehicle (*n* = 9), and TAC + B38-CAP (*n* = 8). All values are means ± SEM. **b–h** Two-way ANOVA with Sidak's multiple-comparisons test. Numbers above square brackets show significant *P*-values.

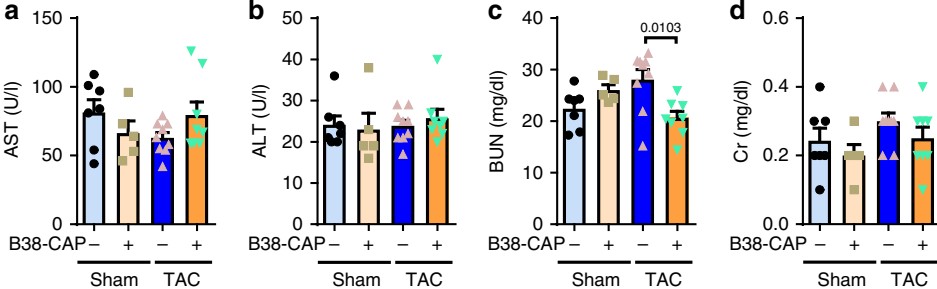

**Fig. 6 No overt toxic effects of B38-CAP on liver and kidney. a**, **b** Liver function test with measurements of aspartate aminotransferase (AST) (**a**) and alanine aminotransferase (ALT) (**b**) in the blood. **c**, **d** Kidney function assessment with measurements of BUN (**c**) and Creatinine (Cr) (**d**) in the blood. sham + vehicle (*n* = 7), sham + B38-CAP (*n* = 5), TAC + vehicle (*n* = 9), and TAC + B38-CAP (*n* = 8). All values are means ± SEM. Two-way ANOVA with Sidak's multiple-comparisons test. Numbers above square brackets show significant *P*-values.

or BA-CAP, were constructed by PCR. PCR products were ligated into a XbaI- and XhoI-double-digested pET28a plasmid, and recombinant proteins were generated by isopropyl β-D-thiogalactopyranoside (IPTG) induction of *E. coli*. Cells were collected and the cell lysate prepared and centrifuged at 13,000 × *g* for 15 min. The resulting supernatant was subjected to ammonium sulfate precipitation, anion-exchange chromatography with a Q-Sepharose Fast Flow column (1.6 × 10 cm; GE Healthcare), and gel filtration chromatography with a Superdex 75 pg column

(2.6 × 60 cm; GE Healthcare) (Supplementary Fig. 2a). To exclude potential contamination of endotoxin, the eluates were further passed through a Polymyxin B column. For preparation of recombinant human ACE2 in Sf9 insect cells, human ACE2 cDNA was inserted into the XbaI and KpnI sites of pFastBac1 vector (Invitrogen) and the generated recombinant bacmid DNA was transfected into Sf9 cells using Cellfectin (Invitrogen) to construct recombinant baculovirus encoding human ACE2. Sf9 cells were infected with the recombinant baculovirus at a

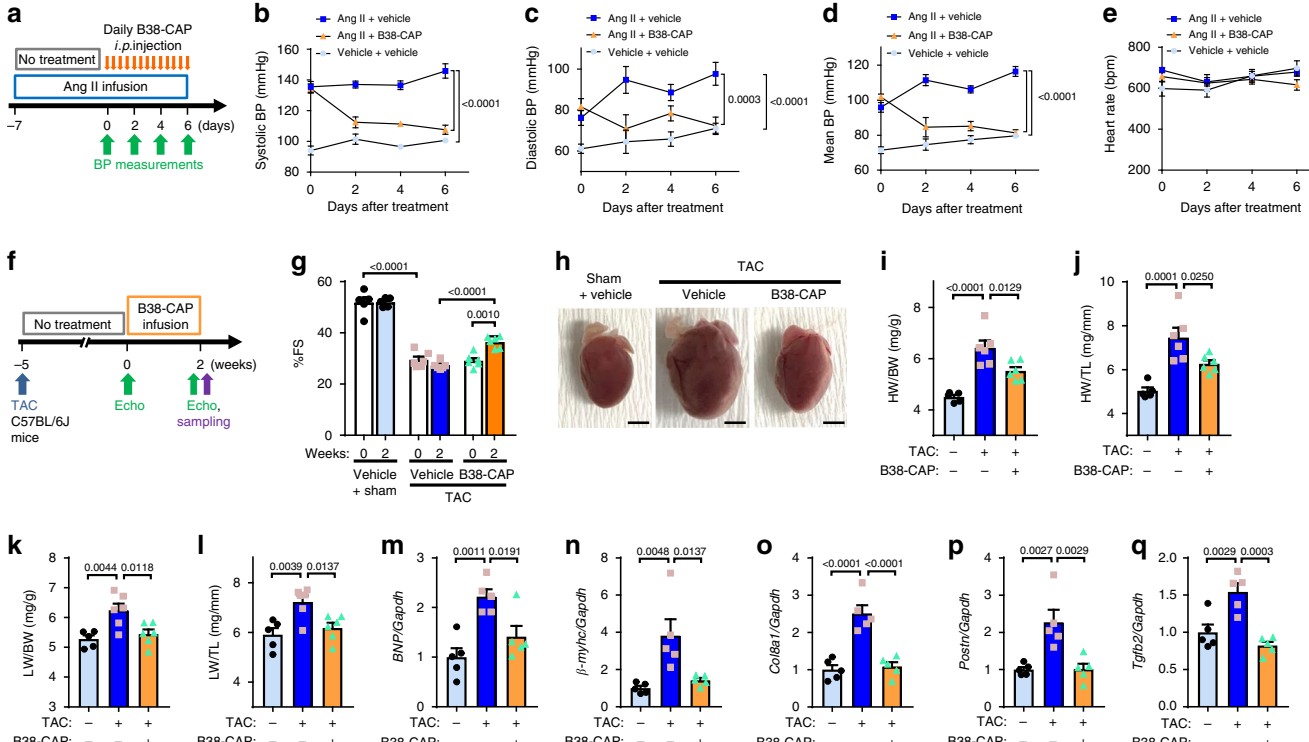

**Fig. 7 Therapeutic effects of B38-CAP on established hypertension and cardiac dysfunction.** a–e Therapeutic effects of B38-CAP on established hypertension. Experimental protocol (**a**); Ang II infusion (1 mg/kg/day) was initiated at 7 days before treatment. The mice were injected with B38-CAP (2 mg/kg i.p.) or vehicle twice a day and blood pressure was measured by tail-cuff system at 2 h after injection. Systolic (**b**), diastolic (**c**), and mean (**d**) BP and heart rate (**e**) in the mice treated Ang II + vehicle (n = 7), Ang II + B38-CAP (n = 7), and vehicle + vehicle (n = 5). f–l Therapeutic effects of B38-CAP on established cardiac dysfunction. Experimental protocol (**f**); the C57BL/6J mice had TAC surgery at 5 weeks before treatment and B38-CAP (2 mg/kg/day) or vehicle was continuously infused with osmotic mini-pumps. Echocardiography parameters of %fractional shortening (%FS) (**g**) in the mice treated with sham + vehicle (n = 5), TAC + vehicle (n = 6), and TAC + B38-CAP (n = 6). Representative photographs of the hearts of mice under TAC (**h**). Bars indicate 2 mm. HW/BW (**i**), HW/TL (**j**), LW/BW (**k**), and LW/TL (**l**) are in the mice treated with sham + vehicle (n = 5), TAC + vehicle (n = 6), and TAC + B38-CAP (n = 6). m–q qRT-PCR analysis for the expression of heart failure genes and pro-fibrosis genes in the hearts (n = 5 mice per group). All values are means ± SEM. **b–e** Two-way ANOVA with Sidak's multiple-comparisons test. **g** One-way ANOVA with Sidak's multiple-comparisons test for comparison of groups. Two-tailed paired t-test between before and after treatment of the same group. **i–q** One-way ANOVA with Sidak's multiple-comparisons test. Numbers next to square brackets show significant P-values.

**Table 3 Echocardiographic parameters in the mice with established cardiac dysfunction treated with B38-CAP for 2 weeks.**

| | Sham + vehicle Before treatment | Sham + vehicle After treatment | TAC + vehicle Before treatment | TAC + vehicle After treatment | TAC + B38-CAP Before treatment | TAC + B38-CAP After treatment |
|---|---|---|---|---|---|---|
| N | 5 | 5 | 6 | 6 | 6 | 6 |
| Age (weeks) | 15 | 17 | 15 | 17 | 15 | 17 |
| BW (g) | 23.14 ± 1.24 | 25.82 ± 1.03 | 23.12 ± 0.87 | 25.92 ± 0.74 | 23.55 ±± 0.83 | 24.30 ± 1.16 |
| HR (bpm) | 594 ± 49 | 631 ± 68 | 551 ± 51 | 559 ± 49 | 570 ± 65 | 644 ± 61 |
| FS (%) | 51.78 ± 4.09 | 51.91 ± 1.56 | 29.49 ± 2.79*** | 27.20 ± 1.43 | 29.06 ± 2.36*** | 36.38 ± 2.03### †† |
| EF (%) | 76.70 ± 3.70 | 83.89 ± 1.27 | 57.44 ± 4.5*** | 53.56 ± 2.28 | 56.70 ± 3.54*** | 67.03 ± 2.86### †† |
| LVESD (mm) | 1.54 ± 0.15 | 1.54 ± 0.13 | 2.43 ± 0.38*** | 2.75 ± 0.14 | 2.49 ± 0.23*** | 2.15 ± 0.17## † |
| LVEDD (mm) | 3.20 ± 0.08 | 3.21 ± 0.22 | 3.44 ± 0.45 | 3.77 ± 0.17 | 3.51 ± 0.34 | 3.38 ± 0.24 |
| IVSD (mm) | 0.74 ± 0.04 | 0.79 ± 0.03 | 1.05 ± 0.17* | 1.04 ± 0.05 | 1.09 ± 0.05*** | 0.99 ± 0.07† |
| PWD (mm) | 0.78 ± 0.06 | 0.89 ± 0.13 | 1.11 ± 0.12*** | 0.99 ± 0.05 | 1.03 ± 0.06*** | 1.08 ± 0.09 |

Results are presented as mean ± SEM. One-way ANOVA plus Sidak's multiple-comparisons test was used to detect significance.
*BW* body weight, *FS* left ventricular fractional shortening, *EF* left ventricular ejection fraction, *HR* heart rate, *IVSD* end-diastolic interventricular septal wall thickness, *LVEDD* left ventricular end-diastolic diameter, *LVESD* left ventricular end-systolic diameter, *PWD* left ventricular end-diastolic posterior wall.
*P < 0.05 vs. sham + vehicle before treatment; **P < 0.001 vs. sham + vehicle before treatment; ***P < 0.0001 vs. sham + vehicle before treatment; #P < 0.05 vs. TAC + vehicle after treatment; ##P < 0.001 vs. TAC + vehicle after treatment; ###P < 0.0001 vs. TAC + vehicle after treatment. Two-tailed paired t-test was used to detect significance. †P < 0.05 vs. TAC + B38-CAP before treatment; ††P < 0.001 vs. TAC + B38-CAP before treatment.

multiplicity of infection (MOI) of 1 pfu/cell and then cultured in SF-900 II serum-free medium (Invitrogen) using 250 mL shaker flasks at 100 r.p.m. at 28 °C for 72 h. rhACE2 was purified with Profinity IMAC Ni-charged resin (Bio-Rad), eluted with 250 mM imidazole, and further dialyzed against phosphate-buffered saline (PBS).

**In vitro ACE2 activity measurements**. For determination of kinetic constants, the $K_m$ and $k_{cat}$ values for B38-CAP and recombinant human ACE2 (Calbiochem) using Nma-His-Pro-Lys(Dnp) as ACE2 substrate were determined by Michaelis–Menten model using GraphPad Prism Version 6.01 (La Jolla, CA, USA)[19,28]. The $IC_{50}$ values of ACE2 inhibitor MLN-4760 (EMD, Millipore) for recombinant human ACE2 and B38-CAP were measured as follows. The reaction mixture contained 40 μl of 0.1 M HEPES pH 7.5, containing 0.3 M NaCl, 0.01% Triton X-100, 0.02% $NaN_3$, 5 μl MLN-4760 solution, and 5 μl of recombinant human ACE2 or B38-CAP in a total volume of 50 μl. The reaction mixture was incubated at 37 °C for 30 min and then the reaction was terminated by adding 0.2 ml of 0.1 M sodium borate buffer pH 10.5. The fluorescein intensity was measured spectrophotometrically at an emission wavelength 440 nm upon excitation wavelength 340 nm (Hitachi F-2500). The sample concentration required to inhibit 50% of B38-CAP activity under the assay condition was taken as the $IC_{50}$ value. For plasma B38-CAP activity measurements, we developed a new B38-CAP-specific substrate Nma-Leu-Pro-Lys(Dnp) by screening the amino acids of P2 position in Nma-X-Pro-Lys(Dnp) substrates (not shown), and the $K_m$, $k_{cat}$, and $k_{cat}/K_m$ values using this substrate were determined to be 9.52 μM, 224 s$^{-1}$, and 23.5 s$^{-1}$ μM$^{-1}$, respectively. Heparinized plasma was diluted with assay buffer and the reaction mixture contained 45 μl of HEPES buffer pH 7.5, 0.3 M NaCl, 20 μM Nma-Leu-Pro-Lys(Dnp), 0.01% Triton X-100, 0.02% $NaN_3$, and 5 μl diluted plasma or recombinant B38-CAP in a total volume of 50 μl. The reaction mixture was incubated at 37 °C for 60 min and then the reaction was terminated by adding 0.2 ml of 0.1 M sodium borate buffer pH 10.5, and the fluorescence intensity was measured. The enzyme concentration in plasma was calculated based on the standard recombinant B38-CAP activity.

**Hydrolysis of angiotensin peptides by B38-CAP**. Each reaction mixture (185.3 μl) was formulated as 1 mg/ml of angiotensin peptides in 15.4 mM HEPES (pH 7.5), 184 mM NaCl, and 2 μg of recombinant B38-CAP, and reactions were incubated at 37 °C. The reaction was terminated by the addition of 14.7 μl of 0.5 M EDTA. The reaction mixture (20 μl) was analyzed by reverse-phase HPLC (TSKgel Super-ODS, 0.46 × 5, or 10 cm, Tosoh Corporation, Tokyo, Japan) and eluted with a linear gradient of 0–100% acetonitrile in 0.05% trifluoroacetic acid (TFA). Quantification was achieved using the peak area of the standard angiotensins I, II, and 1–7 (Peptide Institute, Inc.) and Ang 1–9 (Wako, Osaka, Japan).

**LC-MS quantification of amino acids (Leu, Phe, and His)**. Samples (100 μL) were diluted with 9.2 M perchloric acid (4.3 μL), then centrifuged at $13,000 × g$ for 15 min. A 5 μl aliquot was injected for a Shimadzu LC-MS system (LCMS2020, Kyoto, Japan) equipped with an electrospray ion source with nebulizer gas 1.5 L min$^{-1}$, drying gas 15 L min$^{-1}$, desolvation line temperature 250 °C, and heat block temperature 200 °C. Chromatographic separations were performed with an Intrada Amino Acid column (3 × 100 mm, Imtakt, Kyoto, Japan). Leu and Phe were analyzed with the mobile phase consisting of (mobile phase A) acetonitrile : tetrahydrofuran (THF) : 25 mM ammonium formate : formic acid 10 : 80 : 10 : 0.4 [v/v] and (mobile phase B) acetonitrile : 100 mM ammonium formate 20 : 80 at 0.4 mL min$^{-1}$. The initial mobile-phase composition was 20% B maintained for 4 min, which was gradually increased to 100% B in 7 min, and then maintained at 100% B for 3 min and back to the initial condition of 20% B in 6 min for re-equilibration. Furthermore, His was analyzed with the mobile phase consisting of (mobile phase A) acetonitrile : water : formic acid 85 : 15 : 0.3 [v/v] and (mobile phase B) 100 mM ammonium formate at 0.4 mL min$^{-1}$. The initial mobile-phase composition was 55% B maintained for 4 min, which was gradually increased to 100% B in 8 min, and then maintained at 100% B for 5 min and back to the initial condition of 55% B in 6 min for re-equilibration. Single-ion monitoring in negative mode with $m/z$ 132, 156, and 166, representing Leu, His, and Phe, respectively. Under these conditions, Leu and Phe were eluted at 7.93 and 7.16 min, respectively, and His was eluted at 12.56 min. Quantification was achieved using the peak area of the standard amino acids mixture, type H (Wako, Osaka, Japan).

**Mice**. C57BL/6N or C57BL/6J wild-type male mice were purchased from CLEA Japan, Inc. and maintained at the animal facilities of Akita University Graduate School of Medicine or Research Institute of National Cerebral and Cardiovascular Center. All animal experiments conformed to the Guide for the Care and Use of Laboratory Animals, Eighth Edition, updated by the US National Research Council Committee in 2011, and approvals of the experiments were granted by the ethics review board of Akita University or Research Institute of National Cerebral and Cardiovascular Center. Randomization was performed by using random numbers. In prevention experiments (Figs. 2–6 and Supplementary Figs. 5–9), the mice were assigned by stratified randomization based on BW. In therapeutic experiments (Fig. 7 and Supplementary Fig. 10), the mice were assigned by stratified randomization based on systolic BP for the experiments of Ang II-induced hypertension

(Fig. 7a–e) or %FS for the experiments of TAC-induced cardiac dysfunction (Fig. 7f–q and Supplementary Fig. 10). The treatments and measurements were performed using a double-blind method.

**Invasive measurements of arterial blood pressure**. Acute hemodynamic experiment was performed with 10-week-old male C57BL/6J mice[26,29]. Ninety minutes before hemodynamic measurements, mice were pretreated with an i.p. injection of either B38-CAP (2 mg kg$^{-1}$ i.p.) or sterile PBS as vehicle. Mice were anesthetized with isoflurane (1–1.5%) and body temperature was maintained at 37–38 °C using a heating pad throughout the experiments. The mouse was securely restrained in a supine position and mechanically ventilated with a tracheal cannulation (peak inflation pressure of 10 cm $H_2O$ and 160 breaths min$^{-1}$; Ventelite, Harvard Apparatus). A tapered PE-50 catheter filled with heparinized saline (20 units ml$^{-1}$) was inserted into the right carotid artery to record arterial BP via a pressure transducer. After 5–10 min of stabilization period, systolic, diastolic, and mean BPs were obtained (Power Lab data acquisition system, AD Instruments) and heart rate was calculated using LabChart8 software (AD Instruments). At 90 min after vehicle or B38-CAP injection, Ang II (0.2 mg kg$^{-1}$ i.p.) or vehicle was injected. The changes in BP were analyzed for a subsequent 20 min time period.

**Transverse aortic constriction**. Ten-week-old male C57BL/6N mice were subjected to pressure overload by TAC[30]. In one of the therapeutic experiments (Fig. 7), C57BL/6J mice were used. The same surgical TAC procedure results in more severe dysfunction in C57BL/6N than in C57BL/6J mouse strains. Briefly, mice were anesthetized via i.p. injection of ketamine (100 mg kg$^{-1}$) and xylazine (20 mg kg$^{-1}$), and a longitudinal incision was made in the proximal portion of sternum. The aortic arch was ligated with an overlying 27-gauge needle by 7-0 silk. The needle was immediately removed leaving a discrete region of constriction. The sham-treated group underwent a similar procedure without ligation. Echocardiography was performed at indicated time points after TAC or sham surgery, and mice were then killed by cervical dislocation.

**Echocardiography and blood pressure measurements**. Echocardiographic measurements were performed as previously described[31]. Briefly, conscious mice were gently grabbed in hand or held in the apparatus, echocardiography was performed using Vevo770 equipped with a 30 MHz linear transducer (Visual Sonics). The %FS was calculated as follows: %FS = [(LVEDD – LVESD)/ LVEDD] × 100. M-mode images were obtained for measurement of wall thickness and chamber dimensions with the use of the leading-edge convention adapted by the American Society of Echocardiography. For BP measurements, conscious mice were warmed at 10 min before measurements through during measurements. BP was measured by a programmable sphygmomanometer (BP-200, Softron, Japan) using the tail-cuff method after 5 days of daily training[29].

**Pharmacological intervention**. When we treated the mice with B38-CAP, we examined two routes of administration; daily i.p. injection and subcutaneous continuous infusion with osmotic mini-pumps (Alzet model 1002, Alza Corp.). The dosage and route of B38-CAP and Ang II (Sigma-Aldrich) treatments in the experiments are described in each figure legends. In prevention experiments, treatment with B38-CAP was initiated at the same time as implantation of Ang II-filled osmotic mini-pumps or completion of TAC surgery[30]. For co-infusion of Ang II and B38-CAP into the subcutaneous of mice for 2 weeks (Fig. 2d–h and Fig. 3), Ang II and B38-CAP were loaded into individual pumps and the mice were implanted with the two pumps. When Ang II was infused for 4 weeks (Supplementary Fig. 7), Ang II-filled osmotic mini-pumps (Alzet model 1002) were replaced with a new one at 2 weeks after implantation. Two or 4 weeks after treatment, BP measurement and echocardiography were performed. In the chronic experiments with i.p. injection of B38-CAP, BP was measured at 2 h after i.p. injection (Supplementary Figs. 6 and 7a–e). In therapeutic experiments (Fig. 7 and Supplementary Fig. 10), treatment with B38-CAP was initiated after the establishment of hypertension or pressure overload-induced cardiac dysfunction. To examine the effects of A779 on B38-CAP hypotensive action (Fig. 2m and Supplementary Fig. 9a–d), conscious mice were pre-injected with B38-CAP (2.0 mg kg$^{-1}$ i.p.) or vehicle at 90 min before measurements. Just before injection of Ang II (0.2 mg kg$^{-1}$ i.p.), A779 (0.2 mg kg$^{-1}$ i.p.) or its combination, baseline data of BP was obtained, and then BP was measured every 5 min after injection by a tail-cuff method.

**Histology**. Heart tissues were fixed with 4% formalin and embedded in paraffin. Five-μm-thick sections were prepared and stained with hematoxylin and eosin or Masson's trichrome stain. For measurement of cardiac fibrosis area, the high-resolution images (×100 magnification) of the heart sections stained with Masson's trichrome were taken using a BIOREVO microscope (BZ9000; Keyence) and fibrosis area was quantified using the Image-Pro software (Media Cybernetics).

**Quantitative real-time PCR**. RNA was extracted using TRIzol reagent (Invitrogen) and cDNA synthesized using the PrimeScript RT reagent kit (TAKARA). Sequences of the forward and reverse primers of the genes studied are shown in

Supplementary Table 6. Real-time PCR was run in 96-well plates using a SYBR Premix ExTaq II (TAKARA) according to the instructions of the manufacturer. Relative gene expression levels were quantified by using the Thermal Cycler Dice Real Time System II software (TAKARA).

**Measurements of Ang II and Ang 1–7 in mouse plasma**. Plasma Ang II and Ang 1–7 levels were measured with enzyme-linked immunosorbent assay (ELISA) after peptide extraction[11]. Blood samples were collected in tubes containing ethylene-diamine tetraacetic acid (25 mM), *o*-phenanthroline (0.44 mM), pepstatin A (0.12 mM), and *p*-hydroxymercuribenzoic acid (1 mM), and then centrifuged at $1200 \times g$ for 10 min. The plasma was saved and stored at $-80\,°C$ until further processing. Angiotensin peptides were acidified and subjected to solid-phase extraction using Sep-Pak cartridges (Waters). The quantity of Ang II in the peptide extract was determined using an ELISA kit (Enzo Life Sciences). Plasma Ang 1–7 concentration was also determined by subjecting the peptide extracts to measurements with Ang 1–7 ELISA kit (CUSABIO Biotech).

**Measurements of liver and kidney functions**. Plasma levels of AST or ALT and BUN or Cr were measured as markers of liver and kidney damages, respectively, by using FUJI DRI-CHEM slide kits (Fujifilm Corporation).

**Statistical analyses**. Data are presented as mean values ± SEM. Statistical significance between two experimental groups was determined using Student's two-tailed *t*-test. Comparisons of parameters among three groups were analyzed by one-way analysis of variance (ANOVA), followed by Sidak's multiple-comparisons test. When a comparison is done for groups with two factor levels, two-way ANOVA with Sidak's multiple-comparisons test were used. $P < 0.05$ was considered significant. Power analysis was performed on the results of initial animal experiments by statistical software R. The value of Cohen's *f* was calculated with between-groups variance and within-subgroup variance, the effect size of each experiment data was calculated under $P < 0.05$ and Power 0.8, and the minimal number of animals was determined.

**Reporting summary**. Further information on research design is available in the Nature Research Reporting Summary linked to this article.

## Data availability
The data that support the findings of this study are available from the corresponding author upon reasonable request. Source data used to generate Figs. 1–7 and Supplementary Figs. 3, 5–10 are provided in the Source Data file. The accession codes of DNA or amino acid sequences are listed in Supplementary Table 1. An uncropped image of SDS-PAGE analysis is shown in Supplementary Fig. 11.

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

## Acknowledgements
We thank all members of our laboratories for technical assistance and helpful discussions, and we are grateful to Mrs M. Momma and Dr K. Hiwatashi for assistance in protein preparation and analysis, and to Mrs C. Inoue and Mr T. Takeda for initial animal experiments. K.K. is supported by the Kaken [17H04028] from Japanese Ministry of Science, the Takeda Science Foundation, Uehara Memorial Foundation and Daiichi Sankyo Foundation. Y.I. is supported by the Kaken [17H06179], T.S. is supported by the Kaken [18K15879], and T.Y. is supported by the Kaken [18K15038] from Japanese Ministry of Science.

## Author contributions

S.N., Y.I., S.T., and K.K. conceived the study. T.M., S.N., T.S., T.I., S.Y., T.G., Y.N., J.M.P., H.W., Y.I., S.T., and K.K. designed the methodology and experiments. T.M., S.N., T.S., T. Yamaguchi, M.H., T.I., K.N., T. Yoshihashi, R.O., S.Y., M.N., S.K., T. Yoshiya, K.Y-K., S.M., S.T., and K.K. conducted experiments and/or analyzed data. K.K., T.M., S.N., and T.S. wrote the manuscript with input from all authors.

## Competing interests

The authors declare no competing interests.
