## [Peer Review File · Nature Communications]

Reviewers' comments:

Reviewer #1 (Remarks to the Author):

There are two major claims of the present work. The authors report the first characterization of carboxypeptidase convergent evolution by demonstrating that a bacterial enzyme has activity that is similar to the human ACE2 enzyme. Second, this bacterial enzyme, B38-CAP, is easier to produce and can cleave angiotensin II to angiotensin 1-7 *in vivo*, which can reduce the symptoms of hypertension in mice. I do not believe that the first claim of the paper that "Structural and functional convergent evolution... remained unknown for carboxypeptidases like ACE2", is correct. These carboxypeptidases are metalloenzymes, of which there are already known examples that come from different protein fold-families ('clans' in the MEROPS nomenclature). Further, the ACE2 and B38-CAP enzymes are, in fact, homologs because they are related through divergent evolution. Both enzymes are members of clan MA and the authors themselves show their phylogenetic relationship in Fig 1B. While this relationship is quite distant, independent inspection of the protein fold topology shows the proteins follow the same pattern. Given that hundreds of papers have been published profiling the substrate profile of proteases, the particular profile of this enzyme is only of major interest in light of its potential medical relevance. As an enzymologist and protein engineer by training, I am not qualified to judge the merits of the hypertension work, but believe that if others feel these results are noteworthy, then the work should be accepted with major revisions. The manuscript must be re-written to accurately reflect the correct phylogeny of these enzymes with claims of evolutionary novelty removed or pared back where appropriate.

More nuanced corrections are below by line number:

At numerous points in the text (36, 101, 113, 203, 204, etc.), the authors refer to a concept of "% sequence homology", which should be more accurately stated as sequence identity. Enzymes are either homologous or not; homology cannot be parsed into % in a meaningful way.

100: *Pyrococcus furiosus* is an Archaea, not a Bacteria. Perhaps the authors could use 'microbial' as a more general word?

106-107: The retention of the catalytic pocket throughout divergent evolution speaks to retention of catalytic mechanism, Lewis acid-assisted hydrolysis of a peptide, and not to the sequence specificity of the peptide. The original Lee MM et al paper that reports the BS-CAP structure gives more details on how substrate specificity is encoded.

129: Why is it meaningful that the IC₅₀ values of ACE2 and B38-CAP are the same for Nicotianamine? This small molecule is a Zn chelator and the IC₅₀ doesn't really reflect specific interactions of the enzyme, just that the Zn is labile.

165: %FS needs to be defined at the first point where it is introduced.

212, 221: subtilis (lower case)

215-216: Whether BS-CAP participates in protease-mediated signaling in its native organism is highly speculative. The enzymes likely did not 'converge' on cutting this peptide, they just happen to share enough side-chain and length specificity to have similar activity on this particular peptide. In the absence of additional information, one would assume that BS-CAP cleaves something wholly different in *B. subtilis*.

217-218: What does probiotic supplementation have to do with injecting purified enzymes into the blood stream? This sentence is grossly speculative. The scope of the discussion should remain bounded by the strong data presented by the authors.

254-255: It is appropriate for the authors to discuss the ease of producing the BC-CAP enzyme in *E. coli*. Particularly as cost of production is listed as a major advantage. Therefore, the authors should also include how much protein is obtained per L culture with their procedure. Many enzymes can be expressed in *E. coli* but only yield small amounts to the point where such a route would not be economical.

276: *E. coli* should be italicized.

283-284: It is no longer acceptable practice to use a Lineweaver-Burk plot to determine a k_{cat} or K_M . The authors will obtain more accurate parameters by using a non-linear fitting procedure, which can be implemented Excel or in most standard curve-fitting packages (Prism, Origin, etc.).

343-345: Spaces between numbers and units. Double check throughout.

379: Consistent capitalization of journal titles should be employed throughout.

Figures

Fig 1a – Dark colors on a dark background are difficult to see in print. I suggest swapping to a white background. Further, these are metalloenzymes and the catalytic Zn^{2+} atom needs to be included.

Fig 1b – This figure explicitly shows that ACE2 and B38-CAP are homologs and related through divergent evolution. As described above, this relationship should be accurately conveyed throughout the text.

The font sizes are difficult to read are there is overlap between the text elements and the boxes around some of the graph that is readily fixed.

Reviewer #2 (Remarks to the Author):

1) Complete echocardiography data should be presented for all models (e.g. Ang II and TAC) in supplementary tables i.e. wall thicknesses (PWd, IVSd), dimensions (LVESD, LVEDD) and heart rate. Echo parameters are very dependent on heart rate and based on some of the representative echo images, heart rate is very different e.g. Supp Fig 4d Ang II-vehicle and Ang II- B38-CAP. Some of the

echo images are not sufficiently clear to accurately see the dimensions e.g. Supp Fig 4d: vehicle:vehicle group, Fig 4c: vehicle TAC group.

2) How is the echo data in Fig 3 c and d different to Supp Fig 4 f and g? Are these different subsets of mice? A data point seems to be missing from Fig 3c Ang II- group?

3) For the TAC model I would recommend presenting a table with morphology including body weight, heart weight and lung weight. Did body weight differ between groups before and after TAC and before and after treatment? Body weight can often differ. It is therefore better to present heart weight relative to tibia length as a marker for hypertrophy.

4) Cardiac fibrosis: This should be quantified for both the Ang II and TAC models.

5) Within the checklist it is noted that randomization was performed. What method of randomization was performed?

It is also noted some data has been excluded using Graphpad. The specific data points/mice removed should be stated. It is not sufficient to just remove data points using Grubbs. There are other requirements which must be met.

Reviewer #3 (Remarks to the Author):

In this study by Minato and colleagues, the authors described the discovery of ACE2-like bacterial enzyme. Enzymatic studies show that the enzyme cleaves multiple peptides similar to ACE2. In vivo experiments show a dramatic reduction of blood pressure, hypertrophy, fibrosis and so on. The experiments have not been carried very rigorously, with comparisons made in different situations (eg ACE2 antagonist) and in vivo experiments with very low or unclear number of animals. The authors did not clearly state the gender of their animals in the individual experiments as it is now recommended by most journals.

Table 1: The use of nicotianamine in the resent experiments is a poor choice of inhibitor as the ability of this compound to inhibit ACE2, reported by some of the authors, has not been confirmed by independent groups. In addition, comparison of IC50 and kinetics in that table are not valid since the data reported for ACE2 have been generated with a different and well-accepted antagonist. These experiments should be performed in the same conditions, with a universally recognized ACE2 antagonist.

Figure 1H: The authors' claim that B38-CAP appears to be capable of a 2-step mono carboxyl proteolysis is doubtful. Indeed, according to the bottom panel of Figure 1F, when Ang-1-9 is provided, the only product generated is Ang-1-7, with Phenylalanine released. It is quite strange that no intermediate product (Ang-1-8) is generated and according to the graph provided, no histidine is

released. For the 2-step model to be true, one must assume that 100% of the formed Ang-II is then converted to Ang-1-7, which is not the case in Figure 1E.

Figure 2: the figure legend is not matching the data on the figure. The authors state they used 3-5 mice per group. First, n=3 is very low and an effort should be made to increase that number. Did the authors perform a power analysis to determine the minimal number of animals to reach statistical significance? Second, some of the groups appear to have a lot more than 5 animals. Figure 2b: the plasma levels of Ang-II seem to be quite low considering the massive dose of Ang-II that the authors infused. How can this small increase in plasma Ang-II levels lead to a BP increase of 30 mmHg? Figure 2c: the baseline systolic BP data reported (less than 110 mmHg), using tail-cuff measurement are hard to believe. Conscious mice recorded with telemetry, a stress-free method, have a systolic BP higher than what is reported here. Are those mice anesthetized? Again the number of animals stated in the legend does not match the data presented. Some groups seem to have only 2 animals. It is quite surprising that the Ang-II-mediated hypertension was completely abolished by infusion of the bacterial enzyme. Was there an immune response associated with the chronic infusion of B38-CAP, as it has been reported with rhACE2? How long after IP injection was the BP measured? Since B38-CAP plasma levels are almost gone after 8 hours, BP should be expected to raise again once the enzyme is eliminated.

Figure 3: the representative images of the heart should show a bigger area of the heart, as in Figure 5 so the reader can have a better idea of the amount of fibrosis, and not a selected region.

Figure 5: the representative images do not seem to show much difference in term of fibrosis between the TAC and TAC+B38 groups. This is definitely contrasting with the 6-fold change in collagen presented on the bar graphs. How do the authors explain this discrepancy?

Figure 6: the “toxicity evaluation” data are not very conclusive due to the very small size (n=2) of the sham B38 group. AST and creatinine seem to trend to a major increase and this should be verified by adding a significant number of animals in this group.

Are the beneficial effects of B38-CAP due to a reduction of Ang-II or to the formation of Ang-(1-7)? The authors should include a group of animals infused with Ang-(1-7) or with an Ang-(1-7) antagonist to assess the contribution of the heptapeptide.

The discussion seem to lack focus with the authors putting emphasis on probiotics while nothing in the performed experiments could draw a direct link to that particular use. Notably, B38-CAP was infused or injected IP and nothing indicates that it could be present

Response to the Reviews

Takafumi Minato, et al. "B38-CAP, a bacteria-derived ACE2-like enzyme, suppresses hypertension and cardiac dysfunction" (Manuscript number: NCOMMS-18-33167)

Point-by-point response to the reviewer comments

Reviewer #1

There are two major claims of the present work. The authors report the first characterization of carboxypeptidase convergent evolution by demonstrating that a bacterial enzyme has activity that similar to the human ACE2 enzyme. Second, this bacterial enzyme, B38-CAP, is easier to produce and can cleave angiotensin II to angiotensin 1-7 in vivo, which can reduce the symptoms of hypertension in mice. I do not believe that the first claim of the paper that "Structural and functional convergent evolution... remained unknown for carboxypeptidases like ACE2", is correct. These carboxypeptidases are metalloenzymes, of which there already known examples that come from different protein fold-families ('clans' in the MEROPS nomenclature). Further, the ACE2 and B38-CAP enzymes are, in fact, homologs because they are related through divergent evolution. Both enzymes are members of clan MA and the authors themselves show their phylogenetic relationship is in Fig 1B. While this relationship is quite distant, independent inspection of the protein fold topology shows the proteins follow the same pattern. Given that hundreds of papers have been published profiling the substrate profile of proteases, the particular profile of this enzyme is only of major interest in light of its potential medical relevance. As an enzymologist and protein engineer by training, I am not qualified to judge the merits of the hypertension work, but believe that if others feel these results are noteworthy, then the work should be accepted with major revisions. The manuscript must be re-written to accurately reflect the correct phylogeny of these enzymes with claims of evolutionary novelty removed or pared back where appropriate.

Response: We appreciated your valuable comments on phylogeny of the enzymes. We compared the substrate-binding amino acids of ACE2 with those of BS-CAP and found that the positions of those residues are matched in structure, as mentioned in detail below. We thus agreed with you, and pared back the evolutionary novelty by withdrawing the term "convergent evolution" in the revised manuscript. We state that the bacterial enzymes including B38-CAP are likely homologs of ACE2 with divergent evolution (page 6; lines 21-22). In addition, according to your suggestion, we substantially modified the discussion section and put more emphasis on medical relevance of the B38-CAP enzyme.

More nuanced corrections are below by line number:

At numerous points in the text (36, 101, 113, 203, 204, etc.), the authors refer to a concept of "% sequence homology", which should be more accurately stated as sequence identity.

Enzymes are either homologous or not; homology cannot be parsed into % in a meaningful way.

Response: We replaced “sequence homology” with “sequence identity” throughout the manuscript.

100: Pyrococcus furiosus is an Archea, not a Bacteria. Perhaps the authors could use ‘microbial’ as a more general word?

Response: We replaced it with ‘microbial’.

106-107: The retention of the catalytic pocket throughout divergent evolution speaks to retention of catalytic mechanism, Lewis acid-assisted hydrolysis of a peptide, and not to the sequence specificity of the peptide. The original Lee MM et al paper that reports the BS-CAP structure gives more details on how substrate specificity is encoded.

Response: According to your suggestion, we figured out substrate-binding sites of BS-CAP in the Lee MM et al paper, and then we found that the relevant four amino acids of BS-CAP are positioned in almost the same orientation as those in ACE2 (Arg273/348, His345/234, His505/408 and Tyr515/420 in ACE2/BS-CAP). Please see revised Figure 1a. We also found that the substrate-binding amino acid residues of BS-CAP were conserved in B38-CAP and BA-CAP, and indicated this in revised Supplementary Figure 1.

129: Why is it meaningful that the IC50 values of ACE2 and B38-CAP are the same for Nicotianamine? This small molecule is a Zn chelator and the IC50 doesn’t really reflect specific interactions of the enzyme, just that the Zn is labile.

Response: We conducted the IC50 analysis with MLN-4760, an ACE2 inhibitor, which binds to the catalytic pocket of ACE2.

165: %FS needs to be defined at the first point where it is introduced.

212, 221: subtilis (lower case)

Response: We corrected them.

215-216: Whether BS-CAP participates in protease-mediated signaling in its native organism is highly speculative. The enzymes likely did not ‘converge’ on cutting this peptide, they just happen to share enough side-chain and length specificity to have similar activity on this particular peptide. In the absence of additional information, one would assume that BS-CAP cleaves something wholly different in B. subtilis.

Response: We removed the sentence which describes the potential involvement of BS-CAP in protease-mediated signaling in its native organism *Bacillus subtilis*.

217-218: What does probiotic supplementation have to do with injecting purified enzymes

into the blood stream? This sentence is grossly speculative. The scope of the discussion should remain bounded by the strong data presented by the authors.

Response: We removed the sentences regarding probiotics and added more description on our data in the discussion section.

254-255: It is appropriate for the authors to discuss the easy of producing the BC-CAP enzyme in E. coli. Particularly as cost of production is listed as a major advantage. Therefore, the authors should also include how much protein is obtained per L culture with their procedure. Many enzymes can be expressed in E. coli but only yield small amounts to the point where such a route would not be economical.

Response: According to your suggestion, we conducted new experiments to prepare recombinant human ACE2 protein (rhACE2) in baculovirus-Sf9 insect cell system with His-tag purification and dialysis, and compared the efficiency of protein production of rhACE2 with that of B38-CAP. As a result, the production of recombinant B38-CAP in *E. coli* system (16.8 mg protein yield/culture volume (L)) was more efficient in protein amount than the production of rhACE2 in baculovirus-Sf9 insect cells (5.42 mg protein yield/culture volume (L)), and the time for culture and purification of B38-CAP (2 days) was shorter than that of rhACE2 (6 days, not including baculovirus preparation) (Supplementary Figure 2a-d). It should be noted that B38-CAP was purified with high recovery rate more than 50% (Supplementary Figure 2a) without additional His-tag sequence for purification. Therefore, we conclude that B38-CAP is easily prepared with *E. coli* expression system and cost effective.

276: E. coli should be italicized.

Response: We italicized it.

283-284: It is no longer acceptable practice to use a Lineweaver-Burk plot to determine a kcat or KM. The authors will obtain more accurate parameters by using a non-linear fitting procedure, which can be implemented Excel or in most standard curve-fitting packages (Prism, Origen, etc.).

Response: We re-calculated kcat and Km by using a non-linear fitting procedure with the curve-fitting packages of Prism software.

343-345: Spaces between numbers and units. Double check throughout.

Response: We corrected them and double-checked throughout the text.

379: Consistent capitalization of journal titles should be employed throughout.

Response: We corrected them.

Figures

Fig 1a – Dark colors on a dark background are difficult to see in print. I suggest swapping to a white background. Further, these are metalloenzymes and the catalytic Zn²⁺ atom needs to be included.

Response: We changed the background color to white and added the Zn²⁺ atom in the catalytic site in Figure 1a.

Fig 1b – This figure explicitly shows that ACE2 and B38-CAP are homologs and related through divergent evolution. As described above, this relationship should be accurately conveyed throughout the text.

Response: We modified the description on the relationship of ACE2 and B38-CAP in Figure 1b and state that those bacterial enzymes are likely homologs of ACE2 with divergent evolution (page 6; lines 21-22).

The font sizes are difficult to read are there is overlap between the text elements and the boxes around some of the graph that is readily fixed.

Response: Our apologies for the errors of text elements in figures. We checked and confirmed no errors happened.

Reviewer #2

1) Complete echocardiography data should be presented for all models (e.g. Ang II and TAC) in supplementary tables i.e. wall thicknesses (PWd, IVSd), dimensions (LVESD, LVEDD) and heart rate. Echo parameters are very dependent on heart rate and based on some of the representative ehco images, heart rate is very different e.g. Supp Fig 4d Ang II-vehicle and Ang II- B38-CAP. Some of the echo images are not sufficiently clear to accurately see the dimensions e.g. Supp Fig 4d: vehicle:vehicle group, Fig 4c: vehicle TAC group.

Response: According to your suggestion, we re-conducted all the animal experiments including different conditions, such as higher dosage of Ang II to induce more severe cardiac remodeling. In addition, we also did new experiments of acute hemodynamic measurements. We provided the tables of complete echocardiography data. Please see new Tables 3, 4 and Supplementary Table 3. We also replaced representative echo images with the better quality ones (new Figure 4g).

2) How is the echo data in Fig 3 c and d different to Supp Fig 4 f and g? Are these different subsets of mice? A data point seems to be missing from Fig 3c Ang II- group?

Response: Our sincere apologies for errors in organizing data presentation. We carefully organized the newly obtained data in all figures and also clarified the subsets of mice in each figure legends.

3) For the TAC model I would recommend presenting a table with morphology including body weight, heart weight and lung weight. Did body weight differ between groups before and after TAC and before and after treatment? Body weight can often differ. It is therefore better to present heart weight relative to tibia length as a marker for hypertrophy.

Response: Thank you very much for your valuable comments. We measured body weight before and after TAC, and as pointed out we found that body weight was decreased after TAC. More importantly, the body weight of TAC + B38-CAP group was not decreased after TAC, indicating that general status of the mice of TAC + B38-CAP group was kept better than the mice of TAC + vehicle group (Please see a new Table 4). We also normalized heart weight with tibia length as well as both body weight in the re-conducted experiments (new Figures 3b, 3c, 4c-f; Tables 3, 4; Supplementary Figure 7g, 7h; Supplementary Table 3).

4) Cardiac fibrosis: This should be quantified for both the Ang II and TAC models.

Response: We quantified %fibrosis area of heart tissue sections in the Ang II and TAC models by using the Image-Pro software (new Figures 3h, 5b).

5) Within the checklist it is noted that randomization was performed. What method of randomization was performed?

It is also noted some data has been excluded using Graphpad. The specific data points/mice removed should be stated. It is not sufficient to just remove data points using Grubbs. There are other requirements which must be met.

Response: We performed randomization by dividing mice into groups with equal distribution of body weight, and conducted treatments and measurements in a double-blinded manner. For data exclusion, although we had done Grubbs' outlier analysis for the data of original submission, all of our new data in the revised manuscript were not subjected to outlier analysis. The difference in animal number of TAC + vehicle (n = 9) and TAC + B38-CAP (n = 8) groups (Figures 4-6; Table 4) was due to perioperative death of one mouse at 1 day after TAC surgery which was already assigned to the group of TAC + B38-CAP.

Reviewer #3

In this study by Minato and colleagues, the authors described the discovery of ACE2-like bacterial enzyme. Enzymatic studies show that the enzyme cleaves multiple peptides similar to ACE2. In vivo experiments show a dramatic reduction of blood pressure, hypertrophy, fibrosis and so on. The experiments have not been carried very rigorously, with comparisons made in different situations (eg ACE2 antagonist) and in vivo experiments with very low or

unclear number of animals. The authors did not clearly state the gender of their animals in the individual experiments as it is now recommended by most journals.

Response: According to your suggestion, we re-conducted all the animal experiments more rigorously; We did the experiments with increased animal numbers and/or different conditions, such as higher dosage or long term treatment of Ang II or B38-CAP. In addition, we did new experiments of acute hemodynamic measurements, in parallel with measurements of angiotensin peptides. To further address the mechanisms of the B38-CAP action, we examined Ang 1-7 – Mas receptor axis, a downstream pathway of ACE2, by treating the mice with Ang 1-7 or A779, a Mas receptor antagonist. In addition to completely renewed Figures 2-5, we provided the tables of complete echocardiography data, in which the number and gender of animals are clearly stated. Please see the new Tables 3, 4 and Supplementary table 3. We also replaced representative echo images with the better quality ones (new Figure 4g).

Table 1: The use of nicotianamine in the resent experiments is a poor choice of inhibitor as the ability of this compound to inhibit ACE2, reported by some of the authors, has not been confirmed by independent groups. In addition, comparison of IC50 and kinetics in that table are not valid since the data reported for ACE2 have been generated with a different and well-accepted antagonist. These experiments should be performed in the same conditions, with a universally recognized ACE2 antagonist.

Response: According to your suggestion, to measure IC50 of ACE2 antagonist for B38-CAP, we used MLN-4760, a well-recognized ACE2 antagonist, which binds to the catalytic pocket of ACE2. We also compared the IC50 and kinetics of ACE2 with those of B38-CAP in the same conditions. The results show that B38-CAP exhibits quite similar enzymatic activity to ACE2 in *in vitro* assays. Please see the new Table 1 and Supplementary Figure 3b, 3c.

Figure 1H: The authors' claim that B38-CAP appears to be capable of a 2-step mono carboxyl proteolysis is doubtful. Indeed, according to the bottom panel of Figure 1F, when Ang-1-9 is provided, the only product generated is Ang-1-7, with Phenylalanine released. It is quite strange that no intermediate product (Ang-1-8) is generated and according to the graph provided, no histidine is released. For the 2-step model to be true, one must assume that 100% of the formed Ang-II is then converted to Ang-1-7, which is not the case in Figure 1E.

Response: Thank you for pointing out important aspects of the enzymatic activity of B38-CAP. We noticed that the HPLC analysis cannot detect Histidine and Leucine, which should be released by possible mono-carboxyl proteolysis of Ang 1-9 and Ang 1-10 (Ang I), respectively. We thus first set up a new LC-MS experiment to measure amino acids, Histidine and Leucine as well as Phenylalanine. Then, to determine whether an intermediate product (*e.g.* Ang-1-8 (Ang II)) appears in the proteolysis of Ang I by B38-CAP, we conducted kinetic analysis and measured both peptides and amino acids simultaneously (new Figure 1i, j; Supplementary

Figure 4b). As a result, Ang 1-9, Ang II and Ang 1-7 peptides were detectable at 10 minutes after mixing Ang I with B38-CAP (Figure 1j; Supplementary Figure 4b). Ang 1-7 production increased and finally reached the same levels of the initial Ang I amount, whereas Ang I was undetectable at 90 minutes (Figure 1i, j). On the other hand, Ang 1-9 and Ang II exhibited a minor peak in the middle of reaction, and both peptides became undetectable at the end of the reaction period (Figure 1j). Consistent with peptide kinetics, the amino acids; Leucine (Leu), Histidine (His) and Phenylalanine (Phe) were generated in the same order as mono-carboxyl proteolysis of Ang I, Ang 1-9 and Ang II, respectively (Figure 1j). Therefore, we conclude that the conversion of Ang I to Ang 1-7 by B38-CAP is mediated through three steps of mono-carboxyl proteolysis.

Figure 2: the figure legend is not matching the data on the figure. The authors state they used 3-5 mice per group. First, n=3 is very low and an effort should be made to increase that number. Did the authors perform a power analysis to determine the minimal number of animals to reach statistical significance? Second, some of the groups appear to have a lot more than 5 animals.

Response: Our sincere apologies for errors in organizing data presentation. We conducted a power analysis to determine the minimal number of animals to reach statistical significance by using statistical software R. For the data in original submission, the value of Cohen's f was calculated with between groups variance and within subgroup variance, the effect size of each experiment data was calculated under $P < 0.05$ and Power 0.8, and the minimal number of animals was determined. The results are shown in the table with referenced data figures attached below; the labeling of referenced figures are shown as in the original submission but not in the revised manuscript. Based on these results, we re-performed all the *in vivo* experiments with increased animal number. We then carefully organized the newly obtained data in all figures and also clarified the subsets of mice in each figure legends.

Power analysis to determine the minimal number of animals for appropriate statistics					
1st Experiment	Figures in original submission	Within group variance	Between group variance	Effect size	Required number of samples (n)
					p=0.0001
Plasma AngII	Fig.2b	3404.25	7549.01	1.49	5.67
SBP Day14	Fig.2c	704.98	4832.22	2.62	3.45
MBP Day14	Fig.2e	778.25	4633.05	2.44	3.66
DBP Day14	Fig.2d	1214.62	4613.88	1.95	4.43
AngII HW/BW	Fig.3b	0.91	3.69	2.01	4.23
AngII PWD	Fig.3c	0.17	0.19	1.06	8.71
AngII IVSd	Fig.3d	0.11	0.14	1.14	7.98
SBP Time Course Day14	Supple Fig.4a	690.94	4156.62	2.45	3.6
MBP Time Course Day14	Supple Fig.4b	777.41	4088.49	2.29	3.79
DBP Time Course Day14	Supple Fig.4c	1214.54	4133.05	1.84	4.56
TAC HW/BW	Fig.4b	6.31	24.98	1.99	4.25
TAC PWD	Fig.4d	0.06	0.14	1.53	5.54
TAC LVESD	Fig.4e	1.47	4.38	1.73	4.87
TAC LVEDD	Fig.4f	0.91	1.85	1.43	5.99
TAC %FS	Fig.4g	510.12	1315.76	1.61	5.24
TAC Col8a-1	Fig.5d	58.95	103.97	1.33	6.53
TAC Periostin	Fig.5e	20.59	39.87	1.39	6.14

Power of all experiments are 0.8

Referenced data of initial experiments (figures of original submission)

Figure 2b: the plasma levels of Ang-II seem to be quite low considering the massive dose of Ang-II that the authors infused. How can this small increase in plasma Ang-II levels lead to a BP increase of 30 mmHg?

Response: We re-optimized and improved the conditions of peptide extraction and ELISA assays. We re-measured Ang II levels in the plasma from mice subjected to acute or chronic treatment with Ang II. As a result, acute and chronic treatment with Ang II lead to 40-fold and 4-fold increase of plasma Ang II levels, respectively. Thus, we believe that the increase of BP is consistent with upregulation of plasma Ang II levels.

Figure 2c: the baseline systolic BP data reported (less than 110 mmHg), using tail-cuff measurement are hard to believe. Conscious mice recorded with telemetry, a stress-free method, have a systolic BP higher than what is reported here. Are those mice anesthetized? Again the number of animals stated in the legend does not match the data presented. Some groups seem to have only 2 animals.

Response: In our tail-cuff measurement system, conscious mice exhibit systolic BP less than 110 mmHg, as reported in our previous study (Sato T, et al. J Clin Invest. 2013). We re-performed the BP measurements including different conditions, such as higher dosage of Ang II (1.5 mg/kg/day) and B38-CAP (3 mg/kg/day) (new Figure 2d-h) and obtained the consistent results in terms of hypotensive action of B38-CAP. We believe that our tail-cuff measurement is reliable and reproducible. In addition, for more confirmation, we conducted

invasive hemodynamic analysis to measure the blood pressure in the carotid artery in anesthetized mice using a transducer catheter. Pre-treatment of B38-CAP significantly suppressed Ang II-induced elevation of arterial pressure (new Figure 2b, c; Supplementary Figure 5a-c). All the experiments were performed with more than 6 animals per group.

It is quite surprising that the Ang-II-mediated hypertension was completely abolished by infusion of the bacterial enzyme. Was there an immune response associated with the chronic infusion of B38-CAP, as it has been reported with rhACE2? How long after IP injection was the BP measured? Since B38-CAP plasma levels are almost gone after 8 hours, BP should be expected to raise again once the enzyme is eliminated.

Response: We measured the enzymatic activity of B38-CAP in the plasma of mice during 2 weeks of B38-CAP treatment and found that the activity of continuously infused B38-CAP was detectable in the plasma for 2 weeks (Supplementary Figure 8a, b). In addition, we measured the anti-B38-CAP antibody in the dot blot analysis, which revealed that there is no antibody against B38-CAP detectable in the serum of mice infused with B38-CAP for 2 weeks (Supplementary Figure 8c). For the experiments of IP injection, we measured BP at 2 hours after IP injection. Please see the revised Methods (page 19, line 2) and Figure legends (page 30, lines 12, 19-20) for Supplementary Figures 6 and 7a-e. It should be noted that B38-CAP was detectable though the levels are lower than the high peak at 1-2 hours.

Figure 3: the representative images of the heart should show a bigger area of the heart, as in Figure 5 so the reader can have a better idea of the amount of fibrosis, and not a selected region.

Response: We show a whole area of the heart in Figures 3g, 5a and also quantified %fibrosis area of heart tissue sections by using the Image-Pro software (new Figures 3h, 5b).

Figure 5: the representative images do not seem to show much difference in term of fibrosis between the TAC and TAC+B38 groups. This is definitely contrasting with the 6-fold change in collagen presented on the bar graphs. How do the authors explain this discrepancy?

Response: We re-performed the TAC experiments with increased animal number and obtained the consistent results; 8-fold decrease in %fibrosis area (Figure 5b) and 4-fold decrease in Collagen 8a expression (Figure 5f) in TAC + B38-CAP group compared with TAC group. The discrepancy in the data of original submission may be due to not enough number of animals, and it is solved now.

Figure 6: the “toxicity evaluation” data are not very conclusive due to the very small size (n=2) of the sham B38 group. AST and creatinine seem to trend to a major increase and this should be verified by adding a significant number of animals in this group.

Response: We re-performed the experiments with increased animal number (n = 5-9 per group) and confirmed that B38-CAP does not increase the plasma levels of AST, ALT, BUN and Cre.

Are the beneficial effects of B38-CAP due to a reduction of Ang-II or to the formation of Ang-(1-7)? The authors should include a group of animals infused with Ang-(1-7) or with an Ang-(1-7) antagonist to assess the contribution of the heptapeptide.

Response: We first measured the Ang-(1-7) levels in the plasma of mice treated with Ang II with or without B38-CAP, and observed that Ang-(1-7) levels in the plasma were significantly increased in both acute and chronic experiments (Figure 2k, l). Secondly, when we co-treated the Ang II-injected mice with B38-CAP and A779, an Ang-(1-7) receptor antagonist, the suppressive effects of B38-CAP on Ang II-induced elevation of blood pressure were not affected (Figure 2m; Supplementary Figure 9a-d). Furthermore, co-treatment of Ang-(1-7) and Ang II did not down-regulate Ang II-induced elevation of blood pressure (Supplementary Figure 9e-h). Thus, the hypotensive effects of B38-CAP are provided mainly through downregulation of Ang II levels.

The discussion seem to lack focus with the authors putting emphasis on probiotics while nothing in the performed experiments could draw a direct link to that particular use. Notably, B38-CAP was infused or injected IP and nothing indicates that it could be present

Response: We removed the probiotics from the discussion section, and described more about the effects of IP-injected or infused B38-CAP in mice. B38-CAP was detectable in the plasma of mice either IP-injected or infused with B38-CAP (Figure 2a; Supplementary Figure 8b). In addition, when we measured the blood pressure in the carotid artery in anesthetized mice, we also measured plasma Ang II and Ang 1-7 (Figure 2b-c, 2i and 2k). Massive increase of B38-CAP in the plasma at 90 minutes after IP injection (Figure 2a) significantly downregulated increased Ang II levels while upregulated Ang-(1-7) levels in the plasma (Figure 2i and 2k). The results further support that B38-CAP exists and is functional in the blood of mice.

Reviewers' comments:

Reviewer #1 (Remarks to the Author):

The paper is much improved with the revisions! The authors are to be commended for their hard work.

There are just a couple of small, but important changes to make.

Simply replacing 'homology' with 'identity' is not sufficient. These proteins absolutely have some % sequence identity with each other. They authors have inconsistently written 'no identity' and also 'low identity'. The numbers themselves don't matter in a deep way, but the representation needs to be accurate and consistent throughout.

Recommend for publication.

Reviewer #2 (Remarks to the Author):

-The requested echocardiography data, morphological data, quantification of fibrosis, and clarification regarding exclusions have now been provided. Animals should be randomized with a structured method e.g. computer generated, numbers out of a hat. Any unstructured method has the potential for bias. It sounds like stratified randomization was performed based on body weight? This could be included together with the computer program used.

-Based on the new data and figures with experimental time lines it is now clearer that the treatment was provided immediately after TAC and prior to any cardiac pathology or cardiac dysfunction.

-The ability of this work to be translated and the clinical significance remains unknown because administration of B38-CAP was not given in a setting of established hypertension or cardiac pathology or cardiac dysfunction. This represents a limitation of the study. Numerous agents have been shown to prevent cardiac pathology but had no impact in settings of established disease.

-On page 11, lines 242-243: it is stated that "the decrease of body weight due to TAC heart failure was restored by B38-CAP treatment (Table 4)". This is not strictly true because B38-CAP was administered immediately after the TAC surgery. In this case the treatment prevented a fall in body weight, it did not restore.

Similarly on page 12, lines 249-250: it is stated "We further showed that B38-CAP improves cardiac dysfunction, hypertrophy and fibrosis by pressure overload in mice". To state "improves" there would need to be evidence of cardiac dysfunction, hypertrophy and fibrosis prior to treatment. From what I understand this is only a prevention study based on the timing of treatment. There are other examples of similar statements e.g. Results, page 10, line 211 "B38-CAP reduced Ang II-induced

cardiac fibrosis (Fig 3g, h) and fibrotic genes..". The authors can only conclude that B38-CAP prevented Ang II-induced cardiac fibrosis etc. Introduction page 5, lines 97-98. It is stated that: "We also show beneficial effects of B38-CAP on the pathology of pressure-overload..". This is also not correct, as pathology was not present when the B38-CAP was administered.

This limitation should be included in the discussion.

-For Table 3 and 4 why was one-way ANOVA used rather than two-way when there are 2 factors? Similar comments apply for other figures and tables.

Minor

-In all the tables heart rate should not be presented to 2dp. A measure should only be presented to the accuracy of that measure e.g. 593.67+/- 57.39 should be presented as 594+/- 57.

Reviewer #3 (Remarks to the Author):

NO FURTHER COMMENTS.

Response to referees

Takafumi Minato, et al. "**B38-CAP, a bacteria-derived ACE2-like enzyme, suppresses hypertension and cardiac dysfunction**" (Manuscript number: NCOMMS-18-33167A)

Point-by-point response to the reviewer comments

Reviewer #1

The paper is much improved with the revisions! The authors are to be commended for their hard work. There are just a couple of small, but important changes to make. Simply replacing 'homology' with 'identity' is not sufficient. These proteins absolutely have some % sequence identity with each other. They authors have inconsistently written 'no identity' and also 'low identity'. The numbers themselves don't matter in a deep way, but the representation needs to be accurate and consistent throughout. Recommend for publication.

Response: Thank you very much for your valuable comment. We corrected the inconsistent description on sequence identity, in the abstract section (page 3, lines 8-9) and in the results section (page 6, lines 6-7).

Reviewer #2

-The requested echocardiography data, morphological data, quantification of fibrosis, and clarification regarding exclusions have now been provided. Animals should be randomized with a structured method e.g. computer generated, numbers out of a hat. Any unstructured method has the potential for bias. It sounds like stratified randomization was performed based on body weight? This could be included together with the computer program used.

Response: Thank you for your constructive comments. For most of the experiments to investigate preventive effects of B38-CAP, the mice were assigned by stratified randomization based on body weight. For therapeutic experiments to examine effects of B38-CAP on established disease, the mice were assigned by stratified randomization based on blood pressure or %fractional shortening (%FS) in established hypertension or cardiac dysfunction, respectively. Randomization was performed by giving random numbers to each mice and assigned to each experimental group. By using this method, we excluded potential bias in the results. This is a general and reasonable method employed in clinical statistics. It is not necessary to use a computer software in the experimental settings of laboratory animals, which are already the same in age, gender, health status, strain and genetic background.

-Based on the new data and figures with experimental time lines it is now clearer that the treatment was provided immediately after TAC and prior to any cardiac pathology or cardiac dysfunction.

-The ability of this work to be translated and the clinical significance remains unknown because administration of B38-CAP was not given in a setting of established hypertension or cardiac pathology or cardiac dysfunction. This represents a limitation of the study. Numerous agents have been shown to prevent cardiac pathology but had no impact in settings of established disease.

-On page 11, lines 242-243: it is stated that “the decrease of body weight due to TAC heart failure was restored by B38-CAP treatment (Table 4)”. This is not strictly true because B38-CAP was administered immediately after the TAC surgery. In this case the treatment prevented a fall in body weight, it did not restore.

Similarly on page 12, lines 249-250: it is stated “We further showed that B38-CAP improves cardiac dysfunction, hypertrophy and fibrosis by pressure overload in mice”. To state “improves” there would need to be evidence of cardiac dysfunction, hypertrophy and fibrosis prior to treatment. From what I understand this is only a prevention study based on the timing of treatment. There are other examples of similar statements e.g. Results, page 10, line 211 “B38-CAP reduced Ang II-induced cardiac fibrosis (Fig 3g, h) and fibrotic genes..”. The authors can only conclude that B38-CAP prevented Ang II-induced cardiac fibrosis etc. Introduction page 5, lines 97-98. It is stated that: “We also show beneficial effects of B38-CAP on the pathology of pressure-overload..”. This is also not correct, as pathology was not present when the B38-CAP was administered.

This limitation should be included in the discussion.

Response: According to your suggestion, we first modified the description on the effects of B38-CAP in cardiac dysfunction and pathology in the Results section. Next we conducted new experiments to examine whether B38-CAP has therapeutic effects in established hypertension and cardiac dysfunction. B38-CAP ameliorated established hypertension induced by Ang II infusion (Fig. 7a-d) and improved established cardiac dysfunction induced by TAC pressure overload in two different mouse strains; C57BL/6J and C57BL/6N (Fig. 7e-q; Table 5; Suppl. Fig. 10; Suppl. Table 4). Therefore, we concluded that B38-CAP suppresses hypertension and cardiac dysfunction.

-For Table 3 and 4 why was one-way ANOVA used rather than two-way when there are 2 factors? Similar comments apply for other figures and tables.

Response: We re-analyzed all the data of two factors by using two-way ANOVA.

Minor

-In all the tables heart rate should not be presented to 2dp. A measure should only be presented to the accuracy of that measure e.g. 593.67+/- 57.39 should be presented as 594+/- 57..

Response: We removed the numbers below decimal point from heart rate in all the tables.

REVIEWERS' COMMENTS:

Reviewer #1 (Remarks to the Author):

No further comments.

Reviewer #2 (Remarks to the Author):

I commend the authors on the additional experiments which I think greatly enhance the manuscript.

Below are some minor comments/suggested edits for consideration.

-It is interesting that in the TAC prevention model LV wall thicknesses are smaller with treatment with TAC (Fig 4) but this does not seem to be the case with the reversal models (Table 5 and Supp Table 4) where the impact is mainly on LV dimensions rather than wall thickness. This could be incorporated into the results and/or discussion.

-It would appear that the same surgical TAC procedure results in more severe dysfunction in C57BL/6N vs C57BL/6J. It would be helpful to clearly state this in the methods and to insert "C57BL/6J" on Fig 7f and "C57BL/6N" on Supp Fig 10.

-Fig 7a, it was unclear if B38-CAP was given once daily as suggested by the figure and main text or twice daily as indicated in the legend of Fig 7.

-Supp Fig 7 legend: for consistency with figure 4 and legend change PwD to PWD and IVSd to IVSD.

-Supp Fig 9 legend, line 782: Ang 1-7 instead of A779 to match Supp 9e?

-Supp Fig 10: examples where the legend symbols and fig don't match e.g. panel d. LVESD vs LVDs. Same legend there seems to be duplication of c-e Two-tailed paired t-test.

-In some legends mean +/- sem should be inserted e.g. Fig 1j, Supp Fig 3 b,c

-Supp Table 4: Should be PWD not PWT.

-I recommend you double check that data in Supp Table 4 and Supp Fig 10c match. The SEM are a bit difficult to see but it looks like SEM is larger in TAC B38-CAP at 2 weeks (orange bar- figure) than 0 week (white bar), but this is not the case in the Table. Unless I have misinterpreted something.

Response to the Reviews

Takafumi Minato, et al. "B38-CAP is a bacteria-derived ACE2-like enzyme that suppresses hypertension and cardiac dysfunction" (Manuscript number: NCOMMS-18-33167B)

Point-by-point response to the reviewer comments

Reviewer #2

I commend the authors on the additional experiments which I think greatly enhance the manuscript. Below are some minor comments/suggested edits for consideration.

-It is interesting that in the TAC prevention model LV wall thicknesses are smaller with treatment with TAC (Fig 4) but this does not seem to be the case with the reversal models (Table 5 and Supp Table 4) where the impact is mainly on LV dimensions rather than wall thickness. This could be incorporated into the results and/or discussion.

Response: We added the description that the impact of B38-CAP was on LV dimensions rather than wall thickness (lines 270-271).

-It would appear that the same surgical TAC procedure results in more severe dysfunction in C57BL/6N vs C57BL/6J. It would be helpful to clearly state this in the methods and to insert "C57BL/6J" on Fig 7f and "C57BL/6N" on Supp Fig 10.

Response: We stated that the same surgical TAC procedure results in more severe dysfunction in C57BL/6N than C57BL/6J mouse strains (lines 435-436), and also added "C57BL/6J" and "C57BL/6N" in Fig 7f and Supp Fig 10a, respectively.

-Fig 7a, it was unclear if B38-CAP was given once daily as suggested by the figure and main text or twice daily as indicated in the legend of Fig 7.

Response: We modified Fig 7a to clarify that B38-CAP was given twice daily.

-Supp Fig 7 legend: for consistency with figure 4 and legend change PWd to PWD and IVSd to IVSD.

Response: We changed PWd to PWD and IVSd to IVSD.

-Supp Fig 9 legend, line 782: Ang 1-7 instead of A779 to match Supp 9e?

Response: We corrected A779 to Ang 1-7.

-Supp Fig 10: examples where the legend symbols and fig don't match e.g. panel d. LVESD vs LVDs. Same legend there seems to be duplication of c-e Two-tailed paired t-test.

Response: We corrected LVDs to LVESD in Fig 10d, and also corrected duplication of "Two-tailed paired t-test".

-In some legends mean +/- sem should be inserted e.g. Fig 1j, Supp Fig 3 b,c

Response: We inserted mean +/- sem in Fig 1j and Suppl Fig 3 b,c.

-Supp Table 4: Should be PWD not PWT.

Response: We corrected PWT to PWD.

-I recommend you double check that data in Supp Table 4 and Supp Fig 10c match. The SEM are a bit difficult to see but it looks like SEM is larger in TAC B38-CAP at 2 weeks (orange bar- figure) than 0 week (white bar), but this is not the case in the Table. Unless I have misinterpreted something.

Response: Our apology for error in the graph of Supple Fig 10c. The error bar of TAC B38-CAP at 2 weeks (orange bar) had been mistakenly calculated for SD. We corrected it to SEM.